# CTBench: A Library and Benchmark for Certified Training

Yuhao Mao [1]  Stefan Balauca [2]  Martin Vechev [1]

## Abstract

Training certifiably robust neural networks is an important but challenging task. While many algorithms for (deterministic) certified training have been proposed, they are often evaluated on different training schedules, certification methods, and systematically under-tuned hyperparameters, making it difficult to compare their performance. To address this challenge, we introduce CTBench, a unified library and a high-quality benchmark for certified training that evaluates all algorithms under fair settings and systematically tuned hyperparameters. We show that (1) almost all algorithms in CTBench surpass the corresponding reported performance in literature in the magnitude of algorithmic improvements, thus establishing new state-of-the-art, and (2) the claimed advantage of recent algorithms drops significantly when we enhance the outdated baselines with a fair training schedule, a fair certification method and well-tuned hyperparameters. Based on CTBench, we provide new insights into the current state of certified training, including (1) certified models have less fragmented loss surface, (2) certified models share many mistakes, (3) certified models have more sparse activations, (4) reducing regularization cleverly is crucial for certified training especially for large radii and (5) certified training has the potential to improve out-of-distribution generalization. We are confident that CTBench will serve as a benchmark and testbed for future research in certified training.

## 1. Introduction

As a crucial component of trustworthy artificial intelligence, adversarial robustness (Szegedy et al., 2014; Goodfellow et al., 2015), *i.e.*, resilience to small input perturbations, has established itself as an important research area. While initially the community focused on heuristic methods to craft adversarial examples and defenses against them, it turned out that such defenses are often brittle and can be evaded by adaptive adversaries (Athalye et al., 2018; Tramèr et al., 2020). Thus, neural network certification has emerged as a method for providing provable guarantees on the robustness of a given network (Gehr et al., 2018; Wong & Kolter, 2018; Zhang et al., 2018; Singh et al., 2019).

Two families of neural network certification methods have been proposed: complete methods (Katz et al., 2017; Tjeng et al., 2019) which compute the exact bounds but are extremely computationally expensive, and convex-relaxation based methods (Zhang et al., 2018; Singh et al., 2019) which are more scalable but provide approximate bounds. State-of-the-art (SOTA) verifiers (Xu et al., 2021; Ferrari et al., 2022; Zhang et al., 2022) combine both approaches, by using convex relaxations to speed up the solving of complete methods via Branch-and-Bound (Bunel et al., 2020).

However, the scalability of neural network certification is still a major challenge since the computational complexity of SOTA verifiers grows exponentially with network size. To tackle this issue, certified training (Mirman et al., 2018; Gowal et al., 2018) was proposed to train neural networks that are amenable to certification. Such methods are typically categorized into two groups: (1) training with a sound upper bound of the robust loss (Gowal et al., 2018; Zhang et al., 2020; Shi et al., 2021), and (2) training with an unsound surrogate loss that approximates the exact robust loss (Müller et al., 2023; Mao et al., 2023; De Palma et al., 2024). The latter group has been shown to be more effective.

While certified training has made significant advances, there is currently no benchmark that can be used to fairly evaluate the effectiveness of the different certified training methods. Specifically, the literature often compares against previous methods using quoted numbers due to high computational costs, although the verifier and certification budget differ. These unfair comparisons ultimately hinder the community from drawing reasonable conclusions on the effectiveness of certified training methods. In addition, existing works systematically under-tune hyperparameters, in order to show effectiveness against baselines, thus establishing a weaker SOTA. Further, there is no unified codebase for these meth-

---

[1]Department of Computer Science, ETH Zürich, Switzerland [2]INSAIT, Sofia University "St. Kliment Ohridski", Sofia, Bulgaria. Correspondence to: Yuhao Mao <yuhao.mao@inf.ethz.ch>.

*Proceedings of the 42nd International Conference on Machine Learning*, Vancouver, Canada. PMLR 267, 2025. Copyright 2025 by the author(s).

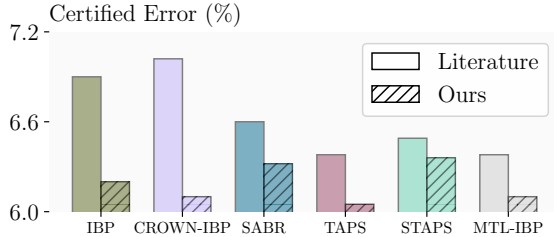

Figure 1: Reduction in certified error on MNIST $\epsilon = 0.3$ (lower is better).

ods, making future development and comparison difficult.

**This work: a Unified Library and High-quality Benchmark for Certified Training**   We address these challenges, for the first time unifying SOTA certified training methods into a single codebase called CTBENCH. This enables a fair comparison between certified training methods and re-establishes a much stronger SOTA by fixing problematic implementations and systematically tuning hyperparameters. As shown in Figure 1, these steps lead to significant improvements uniformly. In addition, we show that the claimed advantage of recent SOTA reduces significantly when we apply the same budget and hyperparameter tuning to all methods. Based on our released model checkpoints, we provide an extensive analysis of the model properties, highlighting many new insights on its loss landscape, mistake patterns, regularization strength, model utilization, and out-of-distribution generalization. We are confident that CTBENCH will serve as a benchmark and testbed for future work in certified training.

## 2. Related Work

We now briefly review works mostly related to ours.

**Benchmarking Certified Robustness**   Li et al. (2023) provides the first benchmark for certified robustness, covering not only deterministic certified training but also randomized certified training and certification methods. However, it is outdated and thus provides little insight into the current SOTA methods. For example, it reports 89% and 51% best certified accuracy for MNIST $\epsilon = 0.3$ and CIFAR-10 $\epsilon = \frac{2}{255}$ in its benchmark, respectively, while recent methods have achieved more than 93% and 62% (Müller et al., 2023; Mao et al., 2023; De Palma et al., 2024).

**Certified Training**   DIFFAI (Mirman et al., 2018) and IBP (Gowal et al., 2018) apply box relaxation to upper bound the worst-case loss for training. Efforts have been made towards applying more precise approximations: Wong et al. (2018) and Balunovic & Vechev (2020) apply DEEPZ

(Singh et al., 2018), and Zhang et al. (2020) incorporate linear relaxations (Zhang et al., 2018; Singh et al., 2019). While these approximations are more precise (Baader et al., 2024; Mao et al., 2025), they often lead to worse training results, attributed to non-smoothness (Lee et al., 2021), discontinuity and sensitivity (Jovanović et al., 2022) of the loss surface. Some recent work (Balauca et al., 2024) aims to mitigate these problems, however, the most effective training approximation is still the least precise box relaxation. In this regard, the focus of the community has shifted towards improving IBP: Shi et al. (2021) propose a new regularization and initialization paradigm to speed up IBP training; De Palma et al. (2022) apply IBP regularization to make adversarial training certifiable; Müller et al. (2023), Mao et al. (2023) and De Palma et al. (2024) propose unsound but more effective IBP-based surrogate losses for training; Mao et al. (2024) propose to use wider models instead of deeper models for IBP-based methods. These methods achieve universal advantages over non-IBP-based methods, and are thus the focus of our work.

## 3. Background

We now introduce the necessary background for our work, both concepts and training algorithms.

### 3.1. Training for Robustness

We present the mathematical notations on adversarial and certified training here. We consider a neural network classifier $f_\theta(x)$ that estimates the log-probability of each class and predicts the class with the highest estimated log-probability.

**Adversarial Training**   A classifier $f_\theta(x)$ is said to be *adversarially robust* with radius $\epsilon$ w.r.t. $L_p$ perturbation if $f_\theta(x + \delta) = y$ for all $\|\delta\|_p \leq \epsilon$, where $y$ is the ground truth label of $x$. Finding an adversarially robust classifier is formally defined to solve a min-max problem $\theta = \arg\min_\theta \mathbb{E}_{x,y} \max_{\|\delta\|_p \leq \epsilon} L(x + \delta)$. In this regard, adversarial training solves the inner maximization problem by generating adversarial examples during training, and the outer minimization problem by optimizing the empirical loss of adversarial examples.

**Certified Training**   A classifier $f_\theta(x)$ is said to be *certifiably robust* if it is adversarially robust and there exists a sound verifier that certifies the robustness. A verifier typically computes an upper bound on the margin $f_i(x + \delta) - f_y(x + \delta)$ and certifies its robustness if the upper bound is negative for all $i \neq y$. Certified training thus replaces the inner maximization problem with an upper bound and minimizes the upper bound during training instead. Since existing certified training algorithms focus solely on $L_\infty$ distance, we only consider $L_\infty$ perturbations

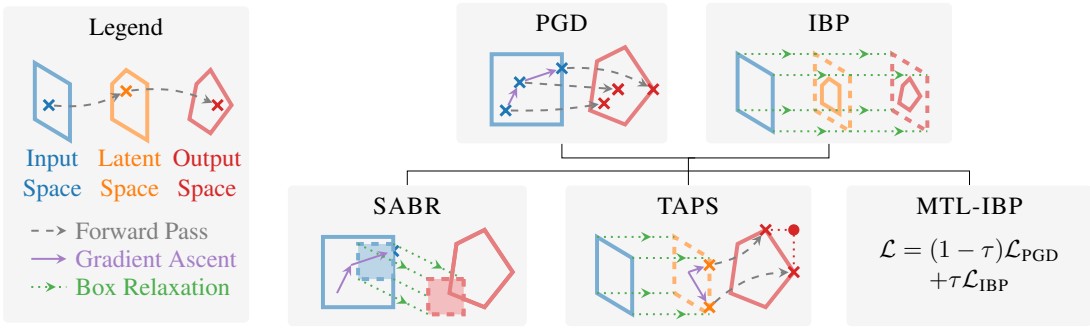

Figure 2: Conceptual overview of core algorithms built into CTBENCH.

in this work and omit the distance type in the notation.

**Metrics** The main metric for certified training is *certified accuracy*, defined to be the ratio of certifiably robust samples in the dataset; *certified error* is defined as one minus the certified accuracy. The ratio of correctly classified samples in the dataset is thus called *natural accuracy*. For reference, we include *adversarial accuracy* as well, defined to be the ratio of adversarially robust samples in the dataset. We apply one of the most widely used SOTA certification methods, MN-BAB (Ferrari et al., 2022), as the verifier. To compute adversarial accuracy, we apply the strong AUTOATTACK (Croce & Hein, 2020) for adversarially trained models, and a combination of PGD attack and branch-and-bound attack from MN-BAB for certifiably trained models. Both attacks have similar empirical strengths, with the latter being slightly stronger on models trained by certified training algorithms due to the completeness of the branch-and-bound attack.

### 3.2. Algorithms in CTBENCH

Now we briefly introduce the core algorithms built into CT-BENCH. Concepts behind them are visualized in Figure 2. A theoretical complexity analysis is provided in Table 11 in App. C.6.

**PGD and EDAC** Projected Gradient Descent (PGD) (Madry et al., 2018) is the most widely recognized adversarial training method. Starting from a randomly initialized point, PGD solves the inner maximization problem by iteratively taking a step towards the gradient ascent direction and clipping the solution into the valid perturbation set. Then, it uses the generated adversarial input $x'$ to lower bound the worst case loss as $L(x')$. Croce & Hein (2020) find that PGD-trained models remains effective against strong attacks, thus it is popular as an integrated part of many certified training methods (Müller et al., 2023; Mao et al., 2023; De Palma et al., 2024). To further improve adversarial robustness, Zhang et al. (2023) improves adversarial generalization via an

extra-gradient method called EDAC, which remains one of the SOTA methods in adversarial training. These methods achieve good but uncertifiable adversarial robustness, hence we use them as adversarial robustness baselines.

**IBP** Interval Bound Propagation (IBP) (Mirman et al., 2018; Gowal et al., 2018) uses interval analysis to approximate the output range of each layer. For example, for the toy network $y = 2 - \text{ReLU}(x_1 + x_2)$ with input bounds $x_1, x_2 \in [-1, 1]$, it first computes the output range of the first layer as $x_1 + x_2 \in [-1, 1] + [-1, 1] \subseteq [-2, 2]$, the second layer as $\text{ReLU}([-2, 2]) \subseteq [0, 2]$ and then final layer as $2 - [0, 2] \subseteq [0, 2]$, thus proving $y \geq 0$ for all possible $x_1, x_2 \in [-1, 1]$. Similarly, IBP computes the layer-wise bounds and then derives an upper bound of the worst-case loss based on the output bounds of the final layer. To stably train models with IBP, Shi et al. (2021) propose to rescale the parameter initialization to ensure constant growth of IBP bounds and a specialized regularization to control the activation status of neurons. They also show that adding a batch norm (Ioffe & Szegedy, 2015) layer before every ReLU layer can improve IBP training. These training tricks are adopted by every IBP-based method introduced below. For brevity, we refer to this variant as IBP in the rest of the paper unless otherwise stated, since it improves the original IBP universally with tricks that facilitate training.

**CROWN-IBP** CROWN-IBP (Zhang et al., 2020) tightens the imprecise interval analysis with linear relaxations of ReLU layers based on IBP bounds and only solves the linear constraints for the final layer output based on CROWN (Zhang et al., 2018), avoiding prohibitive costs during training. To further reduce the cost of solving the bounds for each class, Xu et al. (2020) propose a loss fusion trick to only solve for the final loss, thus reducing the asymptotic complexity by a factor equal to the number of classes. For brevity, we refer to this variant as CROWN-IBP in the rest of the paper unless otherwise stated, since the original CROWN-IBP cannot scale to datasets with many classes, such as TINYIMAGENET.

**SABR** Since IBP is often criticized for the increasingly strong regularization w.r.t. input radius imposed on the neural network, SABR (Müller et al., 2023) proposes to use IBP only for a carefully chosen small box inside the original input box for IBP training. More specifically, it first conducts a PGD attack in the input domain to find an approximately worst-case input, and then takes the surrounding small box with radius $\lambda\epsilon$ around the found input as the input box for IBP training, where $\lambda$ is a pre-defined ratio. For exceptional cases (specifically CIFAR-10 $\epsilon = \frac{2}{255}$), SABR further shrinks the output box of every ReLU towards zero by a pre-defined constant to further reduce the regularization.

**TAPS and STAPS** Observing that IBP relaxation error grows exponentially w.r.t. model depth (Müller et al., 2023; Mao et al., 2024), TAPS (Mao et al., 2023) proposes to split the network into two subparts, using IBP for the first subpart and PGD for the other. This way, the over-approximation from IBP and the under-approximation from PGD partially cancel out, yielding a more precise approximation of the worst-case loss. Further, TAPS uses a separate PGD attack to estimate the bounds of every class to align better with the certification objective. STAPS (Mao et al., 2023) combines TAPS with SABR by using the adversarial small box for TAPS training, thus further reducing regularization.

**MTL-IBP** De Palma et al. (2024) formalizes a family of surrogate loss functions that interpolate between PGD and IBP training. We study MTL-IBP, one of the most effective algorithms in this family. MTL-IBP linearly interpolates between PGD loss and IBP loss, *i.e.*, $\mathcal{L} = (1 - \tau)\mathcal{L}_{\text{PGD}} + \tau\mathcal{L}_{\text{IBP}}$, where $\tau$ is the pre-defined IBP coefficient. To allow more fine-grained control of the interpolation, MTL-IBP uses a larger input radius for the PGD attack for CIFAR-10 when $\epsilon = \frac{2}{255}$.

# 4. A Unified Library and High-quality Benchmark for Certified Training

We now discuss CTBENCH, both the unified library and the corresponding benchmark.

## 4.1. The CTBENCH library

We implement every algorithm described in Section 3.2 in a unified framework. The training loss is composed of three components: the natural loss which measures performance on clean inputs, the robust loss which measures robust performance depending on the concrete algorithms and regularization losses which are used to stabilize training and improve generalization. Formally, the training loss is defined as $\mathcal{L} = (1 - w_{\text{rob}})\mathcal{L}_{\text{nat}} + w_{\text{rob}}\mathcal{L}_{\text{rob}} + \mathcal{L}_{\text{reg}}$. We mainly use $L_1$ regularization to reduce overfitting and the warmup

regularization proposed by Shi et al. (2021) to improve certified training methods. The IBP initialization (Shi et al., 2021) is applied for every certified training method, while adversarial training is initialized with Kaiming uniform (He et al., 2015). Every method has a warmup phase where $\epsilon$ is increased from 0 to the target value and a fine-tuning phase where the model continues to train at the targeted $\epsilon$ to converge. The learning rate is held constant during the warmup phase and decayed twice in the fine-tuning phase with a constant multiplier. We use CNN7 as the model architecture, in agreement with recent literature (Shi et al., 2021; Müller et al., 2023; Mao et al., 2023; De Palma et al., 2024).

Due to the importance of batch norm in certified training, we consider it as a native part of CTBENCH. Specifically, the best practice so far is to set batch norm statistics based on the clean input and use this for computing IBP bounds. However, we find several problematic implementations of batch norm in the literature: (1) when gradient accumulation is involved, the batch norm statistics are not updated correctly, as sub-batch statistics are applied for training; (2) batch norm statistics change more than once before taking a gradient step, as typically the exponentially accumulated statistics are used for conducting a PGD attack and thus evaluating $\mathcal{L}_{\text{rob}}$, while $\mathcal{L}_{\text{nat}}$ is evaluated with batch statistics. The first problem makes gradient accumulation ineffective since the quality of batch statistics depends highly on the batch size, and the second problem prevents training with $w_{\text{rob}} \in (0, 1)$ due to the varied parameters. To address the first problem, we propose to use full batch statistics during gradient accumulation, which leads to slim overheads but allows arbitrary gradient accumulation, as a forward pass is usually much cheaper than a full batch update in certified training. To address the second problem, we conduct PGD attacks with the batch statistics as well and evaluate everything with the current batch statistics. This way, the batch norm statistics are set once per batch just like standard training, allowing training with the combination of $\mathcal{L}_{\text{nat}}$ and $\mathcal{L}_{\text{rob}}$. We remark that the identified problems are systematically ignored in the literature, thus may only be discovered by carefully reading the implementations, which is infeasible for most researchers.

In addition, we find that models trained with the hyperparameters reported in the literature frequently show strong overfitting patterns. To remediate this, we conduct a magnitude search for $L_1$ regularization until the train and validation performance roughly match. To further aid generalization, we apply Stochastic Weight Averaging (Izmailov et al., 2018) for methods that cannot provide metrics for model selection, e.g., MTL-IBP. A more detailed description of the implementation can be found in App. C.

Table 1: CTBENCH results with comparison to the literature. We include the natural accuracy of standard training with CNN7 on each dataset for reference. The best numbers are in bold and those exceeding the literature results are underlined.

| Dataset Std. Nat. [%] | $\epsilon_\infty$ | Training Method | Source | Nat. [%] Literature | Nat. [%] CTBENCH | Cert. [%] Literature | Cert. [%] CTBENCH | Adv. [%] CTBENCH |
|---|---|---|---|---|---|---|---|---|
| | | PGD | / | / | 99.47 | / | $\approx 0^\dagger$ | 98.97 |
| | | EDAC | / | / | 99.58 | / | $\approx 0^\dagger$ | 98.95 |
| | 0.1 | IBP | Shi et al. (2021) | 98.84 | 98.87 | 97.95 | 98.26 | 98.27 |
| | | CROWN-IBP | Xu et al. (2020) | 98.83 | 98.94 | 97.76 | 98.21 | 98.23 |
| | | SABR | Müller et al. (2023) | 99.23 | 99.08 | 98.22 | 98.40 | 98.47 |
| | | TAPS | Mao et al. (2023) | 99.19 | 99.16 | **98.39** | **98.52** | 98.58 |
| | | STAPS | Mao et al. (2023) | 99.15 | 99.11 | 98.37 | 98.47 | 98.50 |
| MNIST | | MTL-IBP | De Palma et al. (2024) | **99.25** | **99.18** | 98.38 | 98.37 | 98.44 |
| 99.50 | | PGD | / | / | 99.43 | / | $\approx 0^\dagger$ | 93.83 |
| | | EDAC | / | / | 99.51 | / | $\approx 0^\dagger$ | 95.02 |
| | 0.3 | IBP | Shi et al. (2021) | 97.67 | 98.54 | 93.10 | 93.80 | 94.30 |
| | | CROWN-IBP | Xu et al. (2020) | 98.18 | 98.48 | 92.98 | 93.90 | 94.29 |
| | | SABR | Müller et al. (2023) | 98.75 | 98.66 | 93.40 | 93.68 | 94.46 |
| | | TAPS | Mao et al. (2023) | 97.94 | 98.56 | **93.62** | **93.95** | 94.66 |
| | | STAPS | Mao et al. (2023) | 98.53 | **98.74** | 93.51 | 93.64 | 94.36 |
| | | MTL-IBP | De Palma et al. (2024) | **98.80** | **98.74** | **93.62** | 93.90 | 94.55 |
| | | PGD | / | / | 88.67 | / | $\approx 0^\dagger$ | 72.41 |
| | | EDAC | / | / | 89.18 | / | $\approx 0^\dagger$ | 72.42 |
| | $\frac{2}{255}$ | IBP | Shi et al. (2021) | 66.84 | 67.49 | 52.85 | 55.99 | 56.10 |
| | | CROWN-IBP | Xu et al. (2020) | 71.52 | 67.60 | 53.97 | 57.11 | 57.28 |
| | | SABR | Müller et al. (2023) | 79.24 | 77.86 | 62.84 | 63.61 | 65.56 |
| | | TAPS | Mao et al. (2023) | 75.09 | 74.44 | 61.56 | 61.27 | 62.62 |
| | | STAPS | Mao et al. (2023) | 79.76 | 77.05 | 62.98 | 64.21 | 66.09 |
| CIFAR-10 | | MTL-IBP | De Palma et al. (2024) | **80.11** | **78.82** | 63.24 | **64.41** | 67.69 |
| 91.27 | | PGD | / | / | 78.71 | / | $\approx 0^\dagger$ | 35.93 |
| | | EDAC | / | / | 78.95 | / | $\approx 0^\dagger$ | 42.48 |
| | $\frac{8}{255}$ | IBP | Shi et al. (2021) | 48.94 | 48.51 | 34.97 | 35.28 | 35.48 |
| | | CROWN-IBP | Xu et al. (2020) | 46.29 | 48.25 | 33.38 | 32.59 | 32.77 |
| | | SABR | Müller et al. (2023) | 52.38 | 52.71 | 35.13 | 35.34 | 36.11 |
| | | TAPS | Mao et al. (2023) | 49.76 | 49.96 | 35.10 | 35.25 | 35.69 |
| | | STAPS | Mao et al. (2023) | 52.82 | 51.49 | 34.65 | 35.11 | 35.54 |
| | | MTL-IBP | De Palma et al. (2024) | **53.35** | **54.28** | **35.44** | **35.41** | 36.02 |
| | | PGD | / | / | 46.78 | / | $\approx 0^\dagger$ | 33.16 |
| | | EDAC | / | / | 46.79 | / | $\approx 0^\dagger$ | 33.16 |
| TINYIMAGENET | $\frac{1}{255}$ | IBP | Shi et al. (2021) | 25.92 | 26.77 | 17.87 | 19.82 | 19.84 |
| 47.96 | | CROWN-IBP | Xu et al. (2020) | 25.62 | 28.44 | 17.93 | 22.14 | 22.31 |
| | | SABR | Müller et al. (2023) | 28.85 | 30.58 | 20.46 | 20.96 | 21.16 |
| | | TAPS | Mao et al. (2023) | 28.34 | 28.64 | 20.82 | 21.58 | 21.71 |
| | | STAPS | Mao et al. (2023) | 28.98 | 30.63 | 22.16 | 22.31 | 22.57 |
| | | MTL-IBP | De Palma et al. (2024) | **37.56** | **35.97** | **26.09** | **27.73** | 28.49 |

† None of the first 10 samples are certified due to the time limit of 1000 seconds per sample.

## 4.2. The CTBENCH benchmark

Table 1 shows the results of CTBENCH using the methodology described in Section 4.1. We further include the average and standard deviation obtained from independent runs in App. D.1, to validate the significance of our results. We find that CTBENCH achieves consistent improvements in certified accuracy for almost all settings, accompanied by increases in natural accuracy in most cases. In particular, it establishes the new SOTA by a margin matching algorithmic advances everywhere except CIFAR-10 $\epsilon = \frac{8}{255}$, where we have 0.03% lower certified accuracy compared to De Palma et al. (2024) but 0.93% higher natural accuracy. This proves the effectiveness of our implementation and the importance of setting batch norm

statistics properly in certified training. We also observe the following: (1) when $\epsilon$ is large, the claimed advantage of recent SOTA over IBP drops significantly, e.g., from $(100-93.10)/(100-93.62)-1 = 8.15\%$ relative certified error reduction to $(100-93.8)/(100-93.95)-1 = 2.48\%$ on MNIST $\epsilon = 0.3$; (2) when the model has sufficient capacity, e.g., on MNIST $\epsilon = 0.1$, certified training can get close to the natural accuracy of standard training (99.18% for MTL-IBP vs 99.50% for standard training), and they also get similar adversarial accuracy to adversarial training (98.58% for TAPS vs 98.95% for EDAC), while certified accuracy is boosted (98.52% for TAPS vs almost 0% for EDAC); (3) when $\epsilon$ is large, certified training even gets better adversarial accuracy than PGD training (94.66% for

TAPS vs 93.83% for PGD on MNIST $\epsilon = 0.3$ and 36.11% for SABR vs 35.93% for PGD on CIFAR-10 $\epsilon = \frac{8}{255}$), but there is still a gap between the adversarial accuracy of the SOTA adversarial training methods and that of the SOTA certified training methods, as well as a similar gap for natural accuracy. We further include a comparison on another architecture, CNN5, between CTBENCH and the implementation of De Palma et al. (2024) in App. D.2, to validate the stability of CTBENCH results across architectures.

## 5. Evaluating and Understanding Certified Models

We now preform an extensive evaluation on models trained with CTBENCH. Our evaluation provides insights into the current state of certified training and addresses several key questions, including the loss fragmentation (Section 5.1), shared mistakes (Section 5.2), model utilization (Section 5.3), regularization strength (Section 5.4), and out-of-distribution generalization (Section 5.5).

### 5.1. Loss Fragmentation

ReLU networks are known to have a fragmented loss surface over the input space, due to the activation switch of neurons. Fragmentation leads to a non-smooth loss surface and increases the difficulty of finding a good approximation of the worst-case loss via gradient-based methods like PGD. Further, SOTA complete certification algorithms relies on branching on different linear regions, and reducing the number of linear regions reduces the certification difficulty. Due to these reasons, in this section, we investigate the fragmentation of loss surfaces in certified models. Specifically, we answer: (1) do certified models have less fragmentation, thus easing adversarial search, and (2) how does the fragmentation change w.r.t. $\epsilon$?

Fragmentation is closely related to the number of unstable neurons, *i.e.*, neurons that switch activation status in the neighborhood, as all fragments are defined by a group of unstable neurons. Vice versa, in most cases, a switching neuron introduces at least one fragmentation since every activation pattern defines a local linear function. Therefore, we can quantify the fragmentation by the ratio of unstable neurons. Since the exact ratio is NP-complete to compute, we use a heuristic but effective method to estimate it: first, a group of inputs is sampled in the input box; second, these inputs are fed into the model to get the corresponding activation pattern; finally, we count the ratio of unstable neurons observed in the sampled activations. This method always establishes a lower bound of the true ratio and gets arbitrarily close when sample size is large enough. In our experiments, we sample the noise 50 times from a standard Gaussian clipped to $[-1, 1]$ and rescale it by $\epsilon$. This sampling focuses more on the neighborhood of the clean input and the

Table 2: Observed count of common mistakes of models on MNIST against their expected values assuming independence across model mistakes.

| | | # models succeeded | | | | | | |
| --- | --- | --- | --- | --- | --- | --- | --- | --- |
| | | 0 | 1 | 2 | 3 | 4 | 5 | 6 |
| $\epsilon = 0.1$ | obs. | 93 | 25 | 21 | 30 | 32 | 56 | 9743 |
| | exp. | 0 | 0 | 0 | 1 | 37 | 900 | 9062 |
| $\epsilon = 0.3$ | obs. | 452 | 73 | 53 | 51 | 80 | 111 | 9180 |
| | exp. | 0 | 0 | 2 | 39 | 445 | 2698 | 6816 |

boundary of the input box, where new fragments appear most likely. We find this sampling process extremely effective; empirically the ratio of unstable neurons observed is very close to the upper bounds derived by IBP for certified models.

Figure 3 shows the result of certified models trained at $\epsilon = 0.1$ and $\epsilon = 0.3$ on MNIST, respectively. We evaluate the fragmentation of every model at both $\epsilon = 0.1$ and $\epsilon = 0.3$. First, we observe that both adversarial training and certified training greatly reduce loss fragmentation compared to standard training, even though many certified training algorithms involve no adversarial attacks. Second, comparing different training methods within each group of □ and ▨, we observe that certified training consistently has significantly less fragmentation than adversarial training when evaluated at the train radius, *e.g.*, IBP reduces fragmentation by 3x compared to EDAC when trained and evaluated at $\epsilon = 0.1$, facilitating the approximation of the worst case loss via adversarial attacks. This is consistent with the practice where a weak single-step attack is adopted in certified training (De Palma et al., 2024), resulting in similar performance as strong attacks but improved efficiency. Third, comparing models trained at different $\epsilon$ (□ vs ▨ and □ vs ▨), we observe that further increasing training $\epsilon$ does not necessarily reduce fragmentation, yet the trend is consistent with adversarial training. These observations prove that certified training can further boost the fragmentation reduction effect of adversarial training, thus introducing more local smoothness into the model. Consistent results on CIFAR-10 are included in App. D.4 as Figure 9.

### 5.2. Shared Mistakes

We now study the correlation between certified models, specifically: do certified models share common mistakes?

We consider models trained by six certified training methods on MNIST at $\epsilon = 0.1$ and $\epsilon = 0.3$ and calculate the distribution of their common mistakes. Specifically, we count the number of models that successfully certify the input, for each sample in the test set containing 10k samples. The observed value is compared with the expected value, defined

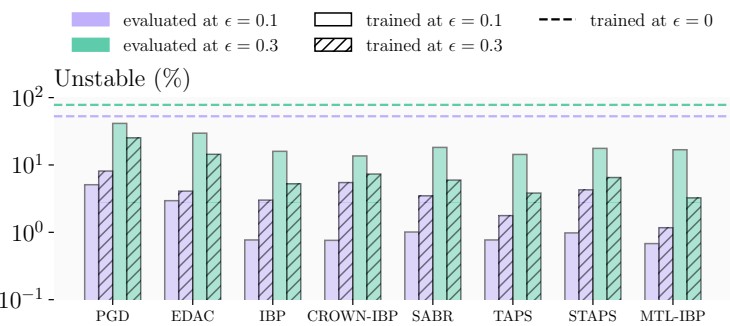

Figure 3: Ratio of unstable neurons for models trained on MNIST with different methods and $\epsilon$.

as expected counts when models with the same certified accuracy make mistakes independently (rounded to the closest integer if necessary), *e.g.*, two models with 90% certified accuracy are expected to have 81% of inputs certified by both. The result is shown in Table 2. Accordingly, certified models share many mistakes, as the number of samples that are certified by none of these models systematically exceeds the expected value. In addition, the number of samples that are certified by all six models is much larger than the corresponding expected value as well. These facts suggest that there could be an intrinsic difficulty score for each input, thus curriculum learning (Bengio et al., 2009; Ionescu et al., 2016) could be a promising direction to improve certified training. This phenomenon is also observed across different certification methods, different model architectures and different datasets, as shown in App. D.3, confirming that this is not a context-specific property but rather an intrinsic property of certified models.

### 5.3. Model Utilization

Model utilization represents how much the model capacity is utilized for the task. IBP is shown to systematically deactivate neurons (Shi et al., 2021) to gain precision. However, it is not yet clear whether more advanced certified training methods deactivate fewer neurons, thus utilizing the model capacity better.

We define model utilization to be the ratio of neurons activated by the clean input. Figure 4 visualizes the result for models trained on MNIST at $\epsilon = 0.1$ and $\epsilon = 0.3$. Surprisingly, we find that more advanced certified training methods, TAPS and MTL-IBP, deactivate more neurons than IBP on MNIST $\epsilon = 0.1$. This is previously believed to be detrimental (Shi et al., 2021), but these models achieve better natural and certified accuracy than IBP. Interestingly, these methods, but not IBP, can retain more utilization on $\epsilon = 0.3$ for better performance where IBP struggles to keep high natural accuracy. Further, we observe that the advanced adversarial training method, EDAC, shows similar behaviors

to TAPS and MTL-IBP, and gets higher adversarial accuracy than PGD. This suggests that the ability to adaptively keep necessary utilization could be crucial to both adversarial and certified robustness. Since dying neurons (Lu et al., 2019) are hard to activate again, future work on better warmup (Shi et al., 2021) could be beneficial, as IBP still struggles to keep necessary model utilization even with the improvements by Shi et al. (2021). More results on CIFAR-10 are included in App. D.4 as Figure 10. There, however, all certified training methods cannot activate more neurons when $\epsilon$ is large, just like IBP in MNIST, while adversarial training methods show similar behavior to MNIST. This suggests that the current certified training methods have not fully solved the utilization problem, especially when $\epsilon$ is large.

### 5.4. Regularization Strength

Previous work (Mao et al., 2024) has shown that IBP bounds are close to optimal bounds for IBP-based certified training, and this condition is established via strong constraints on the model parameters. They quantify this regularization effect by *propagation tightness*, defined to be the ratio between the optimal bound radius and the IBP bound radius, approximating the ReLU network locally with a linear replacement. Intuitively, a close-to-1 propagation tightness means IBP bounds approximately match the exact bounds, and a close-to-0 propagation tightness means IBP bounds are far from the exact bounds. In addition, a high propagation tightness implies strong regularization itself. In this section, we extend the study of propagation tightness to more certified training methods and investigate how it interacts with certified accuracy. Specifically, using propagation tightness as the representative of regularization strength, we answer: (1) does less regularization lead to better certified models, and (2) how does the input radius $\epsilon$ affect this?

Figure 5 shows the interaction between certified accuracy and propagation tightness for certified models trained on MNIST and CIFAR-10. When $\epsilon$ is small (Figure 5a and

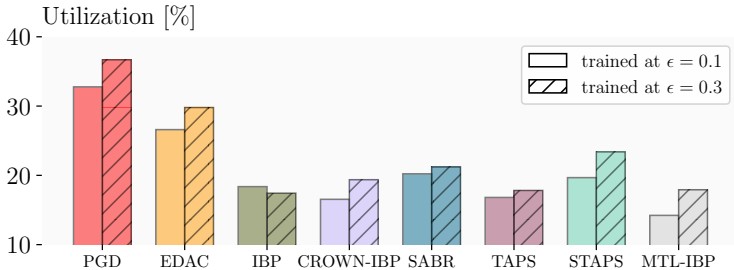

Figure 4: Model utilization for models trained on MNIST with different methods and $\epsilon$. We note that standard training has 42.99% utilization.

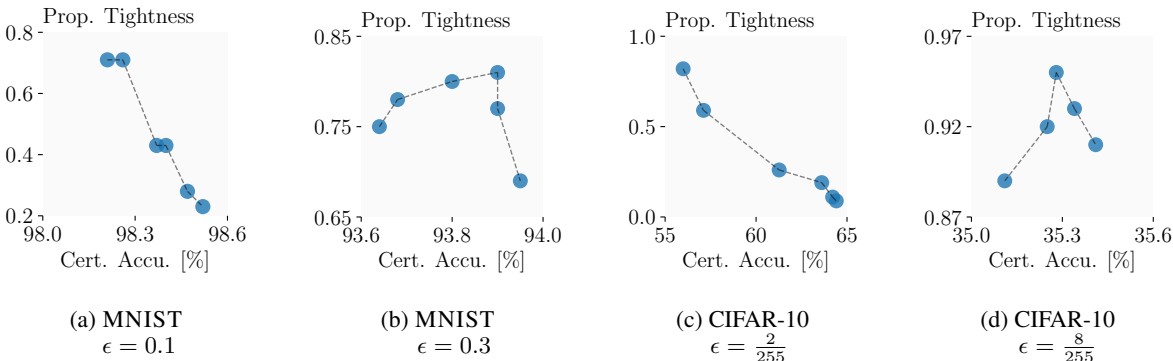

(a) MNIST
$\epsilon = 0.1$

(b) MNIST
$\epsilon = 0.3$

(c) CIFAR-10
$\epsilon = \frac{2}{255}$

(d) CIFAR-10
$\epsilon = \frac{8}{255}$

Figure 5: Certified accuracy vs. propagation tightness for models trained on MNIST and CIFAR-10.

Figure 5c), certified accuracy has a perfectly negative correlation with propagation tightness, *i.e.*, better certified models exhibits less regularization consistently; the best models have close-to-0 propagation tightness. However, when $\epsilon$ is large (Figure 5b and Figure 5d), the correlation is not as clear, and the best model in certified accuracy does not necessarily have the lowest propagation tightness. Instead, models with similar propagation tightness can have significantly different certified accuracy. Nevertheless, models trained for larger radius exhibits much higher propagation tightness. Therefore, we conclude that reducing regularization strength cleverly is crucial for certified training, and the effect is more pronounced when $\epsilon$ is small, while improper reduction could hurt certified accuracy, especially when $\epsilon$ is large. This is consistent with the observation made in Müller et al. (2023) and De Palma et al. (2024) that the best models for small $\epsilon$ often require much less regularization.

### 5.5. Out-of-Distribution Generalization

Out-of-distribution (OOD) generalization is closely related to adversarial robustness (Gilmer et al., 2019). However, the interaction between certified robustness and OOD generalization is not yet clear. We thus investigate the OOD generalization of certified models and answer: (1) do certified models generalize to OOD data, and (2) how does this compare to adversarial training?

We use MNIST-C (Mu & Gilmer, 2019) to evaluate OOD generalization, defined to be the ratio between OOD accuracy and natural accuracy. MNIST-C includes 15 carefully chosen corruptions, covering a broad range of corruptions that are not covered by adversarial robustness while preserving the semantics. We evaluate models trained with both adversarial training and certified training under $\epsilon = 0.1$ and $\epsilon = 0.3$, and report the corresponding OOD accuracy of the model trained via standard training. We note that none of the models has seen these corruptions during training.

Figure 6 depicts the result of OOD generalization for each model on all corruptions. We observe the following: (1) certified training improves OOD generalization compared to standard training except on the *brightness* corruption where both adversarial and certified training fail; (2) certified training shows different OOD generalization patterns to adversarial training, *e.g.*, certified training boosts generalization on the *canny edges* corruption while adversarial training wins on the *stripe* corruption. In general, we find that certified training either greatly boosts the OOD generalization or significantly downgrades the OOD generalization depending on the corruption, and the failure cases are usually those in which adversarial training performs worse than or similarly to standard training. Therefore, we hypothesize that these

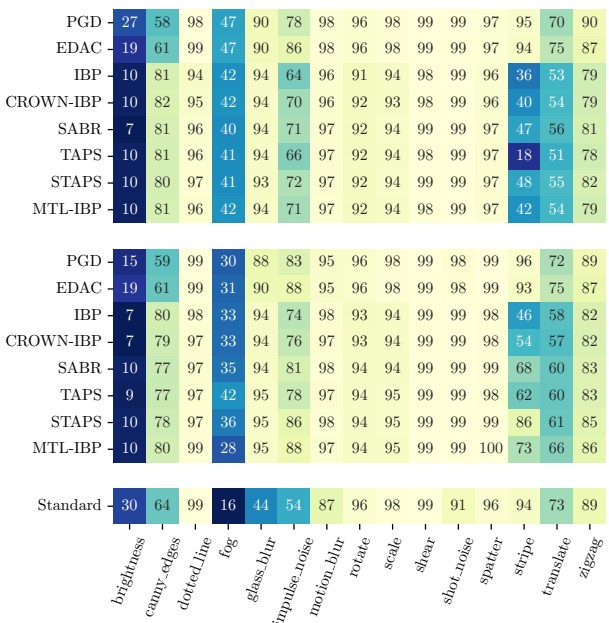

Figure 6: Out-of-distribution generalization evaluated on MNIST-C for models trained on MNIST at $\epsilon = 0.3$ (top), $\epsilon = 0.1$ (middle) and standard training (bottom).

corruptions are at odds with adversarial robustness. Further, different training $\epsilon$ does not significantly affect the OOD generalization except few cases, and ranking in certified accuracy does not show strong correlations with the ranking in OOD generalization. Overall, these results suggest that certified training has the potential to improve OOD generalization to corruptions that standard training struggles with, and the effect is exaggerated when adversarial training improves over standard training. Consistent results on CIFAR-10-C (Hendrycks & Dietterich, 2019) are included in App. D.4 as Figure 11.

## 6. Discussion

**Future Directions** Section 5.1 shows that certified training consistently reduces loss fragmentation, which also benefits adversarial attacks. Therefore, future works may explore architectures and training algorithms that explicitly have little loss fragmentation. In addition, Section 5.2 shows that certified models share mistakes on some hard samples, thus curriculum learning with some well-defined difficulty ranking could facilitate training, where optimization has been known to be particularly hard (Jovanović et al., 2022). Moreover, Section 5.3 shows that even the most advanced certified training method, MTL-IBP, struggles to keep necessary model utilization on large $\epsilon$. Further, Section 5.4 finds that reducing regularization has a different consequence in small and large radius settings; while reducing regularization benefits small radius, it risks decreased performance in

large radius settings. Overall, future work is still required to improve the learning process of certified training. Despite the challenges, Section 5.5 finds that certified models can have surprising and qualitatively different improvement on OOD generalization, which could be a promising application for certified training beyond certified robustness.

**Limitations** We only consider deterministic certified training in CTBENCH, while randomized certified robustness (Cohen et al., 2019) has also made substantial progress. Moreover, we only consider the adversarial robustness, while other types of robustness, such as robustness against patch attacks (Salman et al., 2022) is also important. Finally, we only focus on $L_\infty$ robustness because there exists no *scalable* deterministic certified training algorithm regarding other norms, and leave them as future work.

**Connection to Randomized Certified Training** The issue of unfair comparison highlighted in this work may generalize to the domains of randomized certified robustness, particularly Randomized Smoothing (RS) as introduced by Cohen et al. (2019). Recent RS literature shows considerable difference in evaluation regarding important aspects such as network architectures, training procedures, hyperparameter configurations, and the noise distributions employed for certification. These inconsistencies suggest that unfair comparisons may also be prevalent in RS studies. Therefore, we believe that a similar unified library and benchmark for randomized certified training would be beneficial to the community, allowing for fair comparisons and systematic hyperparameter tuning. We leave this as future work.

## 7. Conclusion

We introduced CTBENCH, a unified library and high-quality benchmark for deterministic certified training on $L_\infty$-norm robustness. Based on CTBENCH, we extensively evaluated certified models trained via state-of-the-art methods, analyzing their regularization strength and utilities. Our analysis reveals that certified training schemes can reduce loss fragmentation, adaptively keep model utilization, make shared mistakes, and generalize well on data with certain corruptions. We are confident that the insights and tools provided by CTBENCH will facilitate future research on certified training and its applications.

## Reproducibility Statement

We release the complete codebase of CTBENCH, including the implementation of all certified training methods and the model checkpoints for the benchmark. The codebase is available at https://github.com/eth-sri/CTBench. A complete description of the experiment setup and hyperparameters is provided in App. C.

## Acknowledgements

This work has been done as part of the EU grant ELSA (European Lighthouse on Secure and Safe AI, grant agreement no. 101070617) and the SERI grant SAFEAI (Certified Safe, Fair and Robust Artificial Intelligence, contract no. MB22.00088). Views and opinions expressed are however those of the authors only and do not necessarily reflect those of the European Union or European Commission. Neither the European Union nor the European Commission can be held responsible for them.

This research was partially funded by the Ministry of Education and Science of Bulgaria (support for INSAIT, part of the Bulgarian National Roadmap for Research Infrastructure).

The work has received funding from the Swiss State Secretariat for Education, Research and Innovation (SERI).

## Impact Statement

This paper presents work whose goal is to advance the field of Machine Learning. There are many potential societal consequences of our work, none of which we feel must be specifically highlighted here.

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

# A. Improvement Decomposition

Decomposition of the universal modifications we made such as batch norm fixes and the hyperparameter tuning is not always possible, as these modifications allow additional vectors of hyperparameter for tuning. For example, we fix batch norm statistics in one batch rather than reset it multiple times as done in some original implementations, allowing $w_{rob}$ to be tuned within $[0, 1]$, while in the literature $w_{rob}$ has to be fixed to 1. Therefore, we cannot formally decompose the effects of hyperparameter tuning and batch norm behaviors, as they are closely dependent on each other.

The literature results are run with three different random seeds, and only the best results among them are reported. This prevents us from substituting our fine-tuned hyperparameter to the original implementation because merely using the same hyperparameters even based on the original implementation hardly reproduces the same number as reported in the literature. In contrast, we run every experiment with the same fixed random seed to allow fair and faithful comparison. Nevertheless, we can showcase the effect for one setting: IBP on MNIST $\epsilon = 0.3$. The literature reports 93.1% certified accuracy, while the same hyperparameter results in 93.18% in our implementation. Further tuning the hyperparameters as in the CTBench benchmark gets 93.8%. While this proves the effectiveness of both the implementation and our hyperparameter tuning, we would like to note that based on previous arguments, this does not faithfully decompose the effect of hyperparameter tuning and batch norm changes, and such decomposition efforts are doomed to fail.

While full disentanglement is infeasible, we conduct preliminary studies to separate implementation advantages from hyperparameter tuning. Table 18 compares CNN5 performance using CTBench and the SOTA codebase, applying CNN7-tuned hyperparameters to both to reduce tuning bias, showing CTBench's universal implementation benefits. Additionally, Table 3 ($L_1$ regularization on IBP-trained networks) and Figure 7 (effects of varying $\lambda$ for SABR and STAPS and $\alpha$ for MTL-IBP on the robustness-accuracy trade-off) illustrate hyperparameter impact.

# B. Additional Discussions

### B.1. Ablation on $L_1$ Regularization

Table 3 shows the effect of $L_1$ regularization on the certified accuracy of IBP-trained networks. We observe that $L_1$ regularization tuned within a small range of hyperparameter choices can improve certified accuracy in most cases, especially for small perturbation sizes.

Table 3: Effect of $L_1$ regularization for IBP-trained networks on different datasets and perturbation sizes.

| Setting | $L_1$ weight | Nat. [%] | Cert. [%] |
|---|---|---|---|
| MNIST 0.1 | 0 | 98.92 | 98.21 |
| | $1 \cdot 10^{-6}$ | 98.84 | 98.22 |
| | $2 \cdot 10^{-6}$ | 98.87 | **98.26** |
| | $5 \cdot 10^{-6}$ | 98.85 | 98.13 |
| MNIST 0.3 | 0 | 98.52 | 93.56 |
| | $1 \cdot 10^{-6}$ | 98.54 | **93.82** |
| | $2 \cdot 10^{-6}$ | 98.51 | 93.66 |
| | $5 \cdot 10^{-6}$ | 98.40 | 93.76 |
| CIFAR 2/255 | 0 | 67.81 | 55.45 |
| | $1 \cdot 10^{-6}$ | 67.49 | **55.99** |
| | $2 \cdot 10^{-6}$ | 66.15 | 55.01 |
| | $5 \cdot 10^{-6}$ | 65.41 | 54.11 |
| CIFAR 8/255 | 0 | 48.51 | **35.28** |
| | $1 \cdot 10^{-6}$ | 48.31 | 34.36 |
| TIN 1/255 | 0 | 25.68 | 19.04 |
| | $2 \cdot 10^{-6}$ | 26.26 | **19.82** |
| | $5 \cdot 10^{-6}$ | 26.37 | 19.80 |
| | $1 \cdot 10^{-5}$ | 26.77 | **19.82** |

### B.2. Robustness-Accuracy Trade-off

The robustness-accuracy trade-off is well-known, where higher certified accuracy often comes at the cost of natural accuracy. Most methods, including SABR and MTL-IBP, have hyperparameters (e.g., $\lambda, \alpha$) that directly regulate this trade-off. Our goal, like in prior work, is to maximize certified accuracy, with natural accuracy improvements seen as a bonus. For completeness, we further provide robustness-accuracy curves, as shown in Figure 7.

Consistent with prior work (Müller et al. (2023, Figure 7) and De Palma et al. (2024, Figure 1)), we observe that reducing regularization initially improves both robustness and natural accuracy, but beyond an optimal point, further reduction severely hurts certifiability while increasing natural accuracy.

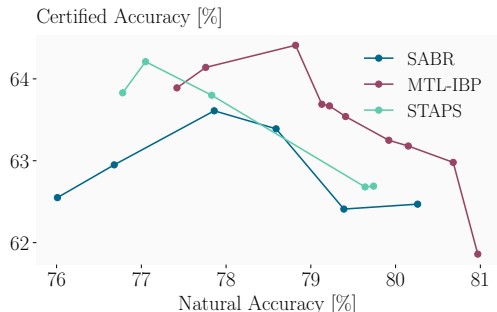

Figure 7: Robustness-Accuracy Trade-off for SABR, STAPS and MTL-IBP on CIFAR-10, $\epsilon = 2/255$.

Additionally, we provide a visualization of how the robustness-accuracy trade-off changes for all certification settings in Figure 8, comparing CTBENCH results with literature results. The transparent points represent previously reported best literature results, while the solid points represent CTBENCH results.

### B.3. Extending $L_\infty$ Deterministic Certified Robustness to Other Norms

Our work focuses on deterministic certified training using bound propagation for the $L_\infty$ norm, as it remains the most reliable and widely adopted approach for robustness guarantees. While Li et al. (2023) explores various norms for certification, it also limits deterministic certified training to $L_\infty$, reflecting the current state of the field, with practical deterministic methods focused on $L_\infty$.

Certification under other norms, such as $L_2$, faces scalability challenges. For example, Wang et al. (2023) evaluate $L_2$ certification on small models with only 192 hidden nodes, while our CNN7 network has over 10M parameters, making their method impractical. Similarly, Soletskyi & Dalrymple (2024) use expensive SDP methods, limiting their approach to synthetic toy datasets (Spheres) and does not naturally extend to $L_\infty$. Furthermore, while Soletskyi & Dalrymple (2024) address $L_2$-norm robustness, their methods do not naturally extend to $L_\infty$.

Bound propagation is difficult for norms other than $L_\infty$. For example, for the $L_2$ norm, it is difficult to track the exact shape of the $L_2$ ball after passing through a linear and ReLU layer. To apply bound propagation, the typical $L_2$ ball with radius $\epsilon = 1$ used in randomized smoothing must be over-approximated by the full $[0, 1]^d$ input set, which guarantees meaningless results. Notably, Wang et al. (2023) do not attempt bound propagation. Developing new deterministic certification methods for other norms is out of the scope of this work.

Exploring deterministic certified training for other norms is a valuable future direction. However, due to scalability limitations and the lack of effective methods for other norms (even on MNIST), our focus remains on $L_\infty$.

### B.4. Comparison with $L_\infty$ Randomized Certified Robustness

In this section, we conduct a preliminary study on comparing $L_\infty$-norm robustness certified by RS to our results based on deterministic algorithms. Specifically, we compare the numbers by the state-of-the-art $L_\infty$-norm RS algorithm of Lyu et al. (2024) on CIFAR-10 at $\epsilon = 2/255$ and $\epsilon = 8/255$ with CTBench results in Table 4. We find that the current RS approaches yield lower certified accuracy compared to CTBench, in agreement with the literature where deterministic methods dominate the $L_\infty$-norm robustness.

### B.5. General Trends Across Datasets

Across the datasets considered in this work, several performance trends emerge, offering insights into how different certification and training methods generalize. For both MNIST and CIFAR-10, we observe that IBP shows decent performance at larger perturbation sizes, while other methods show limited improvement over IBP. This suggests that as perturbations increase in magnitude, stronger regularization is crucial for maintaining certifiability. In the context of corrupted datasets (MNIST-C and CIFAR-10-C), adversarial and certified training methods effectively enhance robustness

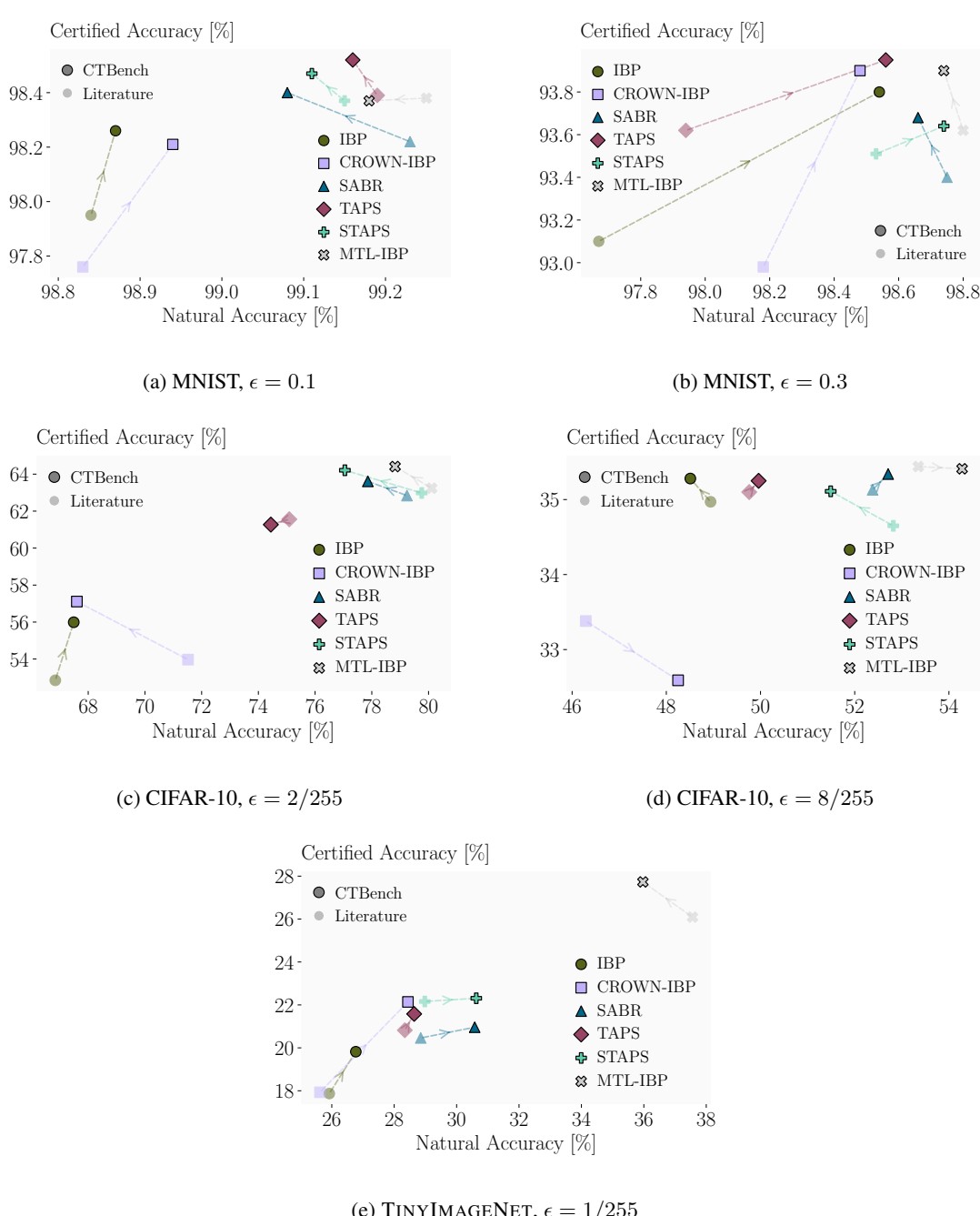

(a) MNIST, $\epsilon = 0.1$

(b) MNIST, $\epsilon = 0.3$

(c) CIFAR-10, $\epsilon = 2/255$

(d) CIFAR-10, $\epsilon = 8/255$

(e) TINYIMAGENET, $\epsilon = 1/255$

Figure 8: Visualization of CTBENCH improvements on the robustness-accuracy trade-off for each certification setting.

Table 4: Comparison between Deterministic Certified Training (this work) and Adaptive Randomised Smoothing (Lyu et al., 2024) on CIFAR-10.

| Method | Nat. [%] | Cert. at $\epsilon = \frac{2}{255}$ [%] | Cert. at $\epsilon = \frac{8}{255}$ [%] |
|---|---|---|---|
| IBP | 67.49 | 55.99 | / |
| IBP | 48.51 | / | 35.28 |
| SABR | 77.86 | 63.61 | / |
| SABR | 52.71 | / | 35.34 |
| MTL-IBP | 78.82 | **64.41** | / |
| MTL-IBP | 54.28 | / | **35.41** |
| ARS $\sigma = 0.12$ | 79.21 | 48.79 | 0.00 |
| ARS $\sigma = 0.25$ | 73.61 | 48.79 | 0.00 |
| ARS $\sigma = 0.50$ | 65.47 | **51.98** | 8.96 |
| ARS $\sigma = 0.75$ | 57.71 | 44.64 | 11.91 |
| ARS $\sigma = 1.00$ | 49.96 | 40.71 | 14.07 |
| ARS $\sigma = 1.50$ | 39.86 | 31.41 | **14.57** |

against localized perturbations such as blur, noise, and pixelation. However, these methods remain less resilient to global transformations like brightness and contrast changes compared to standard training. This observation aligns with the intuition that adversarial and certified training primarily improve robustness in the immediate neighborhood of the original inputs, whereas global changes fall outside this region. More diverse data augmentation strategies may mitigate this problem, though this may also come at the cost of reduced certified adversarial accuracy.

When examining network-level properties such as neuron instability and network utilization, trends across datasets are less straightforward. In all cases, standard training results in the highest neuron instability, as expected due to the absence of regularization aimed at minimizing this effect. However, network utilization does not follow a consistent pattern. In some scenarios, certified training increases network utilization compared to adversarial training, indicating the learning of more complex patterns and relationships. Nevertheless, this trend is not universally observed, suggesting that the underlying dynamics of network utilization may be context-specific.

Overall, these findings highlight that while some performance trends persist across datasets, others are context-dependent, underscoring the need for context-specific analysis when evaluating the current stage of certified robustness methods.

### B.6. Comparing the OOD Generalization of Certified and Adversarial Training

On corrupted datasets (MNIST-C, CIFAR-10-C), adversarial and certified training improve robustness against localized perturbations like blur, noise, and pixelation but struggle with global shifts like brightness and contrast changes. This aligns with the intuition that these methods enhance robustness mainly in the Euclidean neighborhood of the original inputs, whereas global changes fall outside this specification. Moreover, the stronger regularization induced by certified training when compared to adversarial training often exceeds what's needed for untargeted corruptions.

### B.7. Comparing our Loss Fragmentation Result with Shi et al. (2021)

While our approach and the results of Shi et al. (2021) share common concepts and both target to quantify the difficulty of certification, we clarify that Shi et al. (2021) analyze only IBP-bounded instability, which is an over-approximation of the real instability of neurons. In contrast, our analysis applies an estimate of the true number of unstable neurons. To illustrate this difference, we provide a comparison between the two variants in Table 5. We observe that the gap between our lower bound estimate and IBP is larger for SOTA methods, which also reflects in the certification gap between IBP and MN-BAB for these models (Table 22).

Both approaches aimed at quantifying neuron instability are hard to generalize to non-ReLU networks. However, this is of critical importance to the certification of ReLU networks, beyond measuring the difficulty of adversarial attacks which may also be indicated by other smoothness metrics. Concretely, branch-and-bound (BaB), the dominating strategy for complete certification of ReLU networks, directly branches the unstable neurons; thus, the number of unstable neurons provides a direct metric for the difficulty of certification. Since ReLU networks dominate certified training, we adopt the number of unstable neurons as the main metric.

Table 5: Unstable neuron estimates using our sampling method (lower bound) and IBP (upper bound) for networks trained on CIFAR-10 $\epsilon = 2/255$.

| Method | Unstable Neurons [%] | |
| --- | --- | --- |
| | lower bound | upper bound (IBP) |
| IBP | 0.97 | 1.26 |
| CROWN-IBP | 1.16 | 1.69 |
| SABR | 1.37 | 2.82 |
| TAPS | 1.00 | 1.68 |
| STAPS | 1.32 | 2.64 |
| MTL-IBP | 1.53 | 3.42 |

# C. Experiment Details

## C.1. Dataset

We use the MNIST (LeCun et al., 2010), CIFAR-10 (Krizhevsky et al., 2009) and TINYIMAGENET (Le & Yang, 2015) datasets for our experiments. All are open-source and freely available with unspecified license. The data preprocessing mostly follows De Palma et al. (2024). For MNIST, we do not apply any preprocessing. For CIFAR-10 and TINYIMAGENET, we normalize with the dataset mean and standard deviation and augment with random horizontal flips. We apply random cropping to $32 \times 32$ after applying a 2 pixel zero padding at every margin for CIFAR-10, and random cropping to $64 \times 64$ after applying a 4 pixel zero padding at every margin for TINYIMAGENET. We train on the corresponding train set and certify on the validation set, as adopted in the literature (Shi et al., 2021; Müller et al., 2023; Mao et al., 2023; De Palma et al., 2024).

## C.2. Model Architectures

We follow Shi et al. (2021); Müller et al. (2023) and use a CNN7 with Batch Norm for our main experiments. CNN7 is a convolutional network with 7 convolutional and linear layers. All but the last linear layer are followed by a Batch Norm and ReLU layer. This architecture is found to achieve uniformly better results across settings (Shi et al., 2021), and thus is adopted by the literature (Shi et al., 2021; Müller et al., 2023; Mao et al., 2023; De Palma et al., 2024). For TINYIMAGENET, the stride of the last convolution is doubled to reduce the cost.

## C.3. Training Details

**Initialization** Adversarial training methods are initialized by Kaiming uniform (He et al., 2015), while certified training methods are initialized by IBP initialization (Shi et al., 2021).

**Training Schedule** We mostly follow the training schedule of (De Palma et al., 2024), but in some cases a shorter schedule to reduce cost. Specifically, the warmup phase is 20 epochs for MNIST $\epsilon = 0.1$ and $\epsilon = 0.3$, 80 epochs for CIFAR-10 $\epsilon = \frac{2}{255}$, 120 epochs for CIFAR-10 $\epsilon = \frac{8}{255}$ and 80 epochs for TINYIMAGENET $\epsilon = \frac{1}{255}$. In addition, for CIFAR-10 and TINYIMAGENET, we use standard training for 1 additional epoch at the beginning. We apply the IBP regularization proposed by (Shi et al., 2021), with weight equals 0.5 on MNIST and CIFAR-10, and 0.2 on TINYIMAGENET, during the warmup phase. In total, we train 70 epochs for MNIST $\epsilon = 0.1$ and $\epsilon = 0.3$, 160 epochs for CIFAR-10 $\epsilon = \frac{2}{255}$, 240 epochs for CIFAR-10 $\epsilon = \frac{8}{255}$, and 160 epochs for TINYIMAGENET $\epsilon = \frac{1}{255}$.

**Optimization** We use Adam (Kingma & Ba, 2015) with a learning rate of 0.0005. The learning rate is decayed by a factor of 0.2 at epoch 50 and 60 for MNIST $\epsilon = 0.1$ and $\epsilon = 0.3$, at epoch 120 and 140 for CIFAR-10 $\epsilon = \frac{2}{255}$, at epoch 200 and 220 for CIFAR-10 $\epsilon = \frac{8}{255}$, and at epoch 120 and 140 for TINYIMAGENET $\epsilon = \frac{1}{255}$. We use a batch size of 256 for MNIST, and 128 for CIFAR-10 and TINYIMAGENET. Gradients of each step are clipped to 10 in $L_2$ norm. No weight decay is applied and $L_1$ regularization only on weights of linear and convolution layers is used. Further, Wu & Johnson (2021) find that running statistics lag behind the population statistics and propose to use the population statistics for testing. We adopt this strategy in CTBENCH, since it only needs to compute $\mathcal{L}_{\text{nat}}$ and is much cheaper than the computation of $\mathcal{L}_{\text{rob}}$.

Table 6: Best hyperparameter for MNIST $\epsilon = 0.1$.

| | PGD | EDAC | IBP | CROWN-IBP | SABR | TAPS | STAPS | MTL-IBP |
|---|---|---|---|---|---|---|---|---|
| $L_1$ regularization | $1 \times 10^{-5}$ | $1 \times 10^{-5}$ | $2 \times 10^{-6}$ | $2 \times 10^{-6}$ | $1 \times 10^{-6}$ | $1 \times 10^{-6}$ | $1 \times 10^{-6}$ | $1 \times 10^{-5}$ |
| $w_{\mathrm{rob}}$ | 1.0 | 1.0 | 1.0 | 1.0 | 0.7 | 0.7 | 0.7 | 0.7 |
| Train $\epsilon$ | 0.2 | 0.2 | 0.2 | 0.2 | 0.2 | 0.2 | 0.2 | 0.2 |
| $\epsilon$ shrink ratio | / | / | / | / | 0.4 | / | 0.4 | / |
| Classifier size | / | / | / | / | / | 3 | 1 | / |
| TAPS gradient scale | / | / | / | / | / | 4 | 4 | / |
| ReLU shrink ratio | / | / | / | / | / | / | / | / |
| IBP coefficient | / | / | / | / | / | / | / | 0.02 |

## C.4. Tuning Scheme

We conduct a hyperparameter tuning for each method to ensure the best performance, and reduce the search space whenever appropriate based on human knowledge. The search space for each hyperparameter is as follows:

- *$L_1$ regularization*: $\{1 \times 10^{-6}, 2 \times 10^{-6}, 5 \times 10^{-6}, 1 \times 10^{-5}, 2 \times 10^{-5}, 5 \times 10^{-5}\}$. We include $3 \times 10^{-6}$ specifically for CIFAR-10 $\epsilon = \frac{2}{255}$, as this is the value reported by De Palma et al. (2024).

- *$w_{rob}$*: $\{0.7, 0.8, 0.9, 1.0\}$. Surprisingly, $w_{\mathrm{rob}}$ not equal to $1$ can improve both certified and natural accuracy by a large margin when $\epsilon$ is small.

- *Train $\epsilon$*: we use 2x train $\epsilon$ for MNIST $\epsilon = 0.1$, and tune within $\{1x, 1.25x, 1.5x\}$ specifically for CIFAR-10 $\epsilon = \frac{2}{255}$. For others, we use the test $\epsilon$ for training.

- *$\epsilon$ shrink ratio for* SABR *and* STAPS: we mostly keep the value in the literature. When we observe large certifiability gap, we increase the shrink ratio by $0.1$ until the performance fails to increase consistently.

- *Classifier size for* TAPS *and* STAPS: we keep the value in the literature for TAPS, and include only 1 ReLU layer in the classifier for STAPS universally.

- TAPS *gradient scale*: $\{1, 2, 3, 4, 6, 8\}$.

- *ReLU shrink ratio for* SABR *and* STAPS: we keep the value in the literature, thus shrinking the output box of each ReLU by multiplying $0.8$ on CIFAR-10 $\epsilon = \frac{2}{255}$ and do not apply this in other settings.

- IBP *coefficient for* MTL-IBP: $\{0.01, 0.02, 0.05\}$ for MNIST $\epsilon = 0.1$, CIFAR-10 $\epsilon = \frac{2}{255}$ and TINYIMAGENET $\epsilon = \frac{1}{255}$, and $\{0.4, 0.5, 0.6\}$ for MNIST $\epsilon = 0.3$, CIFAR-10 $\epsilon = \frac{8}{255}$.

- *Attack Strength*: we use 3 restarts everywhere for the attack. By default, we use 10 steps for MNIST $\epsilon = 0.1$, 5 steps for MNIST $\epsilon = 0.3$, 8 steps for CIFAR-10 $\epsilon = \frac{2}{255}$, 10 steps for CIFAR-10 $\epsilon = \frac{8}{255}$, and 1 step for TINYIMAGENET $\epsilon = \frac{1}{255}$. However, we find MTL-IBP benefits from using only 1 step everywhere, while more steps will hurt certified accuracy, thus we only use 1 step specifically for MTL-IBP except CIFAR-10 $\epsilon = \frac{2}{255}$, consistent to De Palma et al. (2024). We further only use 2x attack $\epsilon$ for MTL-IBP on CIFAR-10 $\epsilon = \frac{2}{255}$.

We report the best hyperparameter for each method respectively in Table 6, Table 7, Table 8, Table 9, and Table 10.

## C.5. Certification Details

We combine IBP (Gowal et al., 2018), CROWN-IBP (Zhang et al., 2020), and MN-BAB (Ferrari et al., 2022) for certification running the most precise but also computationally costly MN-BAB only on samples not certified by the other methods. The timout for each input is set to 1000 seconds.

Table 7: Best hyperparameter for MNIST $\epsilon = 0.3$.

|  | PGD | EDAC | IBP | CROWN-IBP | SABR | TAPS | STAPS | MTL-IBP |
|---|---|---|---|---|---|---|---|---|
| $L_1$ regularization | $5 \times 10^{-6}$ | $5 \times 10^{-6}$ | $1 \times 10^{-6}$ | $1 \times 10^{-6}$ | $2 \times 10^{-6}$ | $2 \times 10^{-6}$ | $2 \times 10^{-6}$ | $1 \times 10^{-6}$ |
| $w_{\text{rob}}$ | 1.0 | 1.0 | 1.0 | 1.0 | 1.0 | 1.0 | 1.0 | 1.0 |
| Train $\epsilon$ | 0.3 | 0.3 | 0.3 | 0.3 | 0.3 | 0.3 | 0.3 | 0.3 |
| $\epsilon$ shrink ratio | / | / | / | / | 0.8 | / | 0.8 | / |
| Classifier size | / | / | / | / | / | 1 | 1 | / |
| TAPS gradient scale | / | / | / | / | / | 3 | 1 | / |
| ReLU shrink ratio | / | / | / | / | / | / | / | / |
| IBP coefficient | / | / | / | / | / | / | / | 0.5 |

Table 8: Best hyperparameter for CIFAR-10 $\epsilon = 2/255$.

|  | PGD | EDAC | IBP | CROWN-IBP | SABR | TAPS | STAPS | MTL-IBP |
|---|---|---|---|---|---|---|---|---|
| $L_1$ regularization | $2 \times 10^{-5}$ | $5 \times 10^{-6}$ | $1 \times 10^{-6}$ | $1 \times 10^{-6}$ | $1 \times 10^{-6}$ | $2 \times 10^{-6}$ | $5 \times 10^{-6}$ | $3 \times 10^{-6}$ |
| $w_{\text{rob}}$ | 1.0 | 1.0 | 1.0 | 1.0 | 0.7 | 1.0 | 1.0 | 0.9 |
| Train $\epsilon$ | 2/255 | 2/255 | 2/255 | 2/255 | 3/255 | 2/255 | 3/255 | 2/255 |
| $\epsilon$ shrink ratio | / | / | / | / | 0.1 | / | 0.1 | / |
| Classifier size | / | / | / | / | / | 5 | 1 | / |
| TAPS gradient scale | / | / | / | / | / | 5 | 5 | / |
| ReLU shrink ratio | / | / | / | / | 0.8 | / | 0.8 | / |
| IBP coefficient | / | / | / | / | / | / | / | 0.01 |

Table 9: Best hyperparameter for CIFAR-10 $\epsilon = 8/255$.

|  | PGD | EDAC | IBP | CROWN-IBP | SABR | TAPS | STAPS | MTL-IBP |
|---|---|---|---|---|---|---|---|---|
| $L_1$ regularization | $1 \times 10^{-6}$ | $1 \times 10^{-6}$ | 0 | 0 | 0 | 0 | 0 | 0 |
| $w_{\text{rob}}$ | 1.0 | 1.0 | 1.0 | 1.0 | 1.0 | 1.0 | 1.0 | 1.0 |
| Train $\epsilon$ | 8/255 | 8/255 | 8/255 | 8/255 | 8/255 | 8/255 | 8/255 | 8/255 |
| $\epsilon$ shrink ratio | / | / | / | / | 0.7 | / | 0.9 | / |
| Classifier size | / | / | / | / | / | 1 | 1 | / |
| TAPS gradient scale | / | / | / | / | / | 2 | 2 | / |
| ReLU shrink ratio | / | / | / | / | / | / | / | / |
| IBP coefficient | / | / | / | / | / | / | / | 0.5 |

Table 10: Best hyperparameter for TINYIMAGENET $\epsilon = 1/255$.

|  | PGD | EDAC | IBP | CROWN-IBP | SABR | TAPS | STAPS | MTL-IBP |
|---|---|---|---|---|---|---|---|---|
| $L_1$ regularization | $5 \times 10^{-5}$ | $1 \times 10^{-5}$ | $1 \times 10^{-5}$ | $1 \times 10^{-5}$ | $1 \times 10^{-5}$ | $1 \times 10^{-5}$ | $1 \times 10^{-5}$ | $5 \times 10^{-5}$ |
| $w_{\text{rob}}$ | 1.0 | 1.0 | 1.0 | 1.0 | 1.0 | 1.0 | 1.0 | 0.7 |
| Train $\epsilon$ | 1/255 | 1/255 | 1/255 | 1/255 | 1/255 | 1/255 | 1/255 | 1/255 |
| $\epsilon$ shrink ratio | / | / | / | / | 0.4 | / | 0.6 | / |
| Classifier size | / | / | / | / | / | 1 | 1 | / |
| TAPS gradient scale | / | / | / | / | / | 8 | 4 | / |
| ReLU shrink ratio | / | / | / | / | / | / | / | / |
| IBP coefficient | / | / | / | / | / | / | / | 0.05 |

## C.6. Computation

We train and certify MNIST $\epsilon = 0.1$, MNIST $\epsilon = 0.3$ and CIFAR-10 $\epsilon = \frac{8}{255}$ models on a single NVIDIA GeForce RTX 2080 Ti with Intel(R) Xeon(R) Silver 4214R CPU @ 2.40GHz and 530GB RAM. We train and certify CIFAR-10 $\epsilon = \frac{2}{255}$ and TINYIMAGENET $\epsilon = \frac{1}{255}$ models on a single NVIDIA L4 with Intel(R) Xeon(R) CPU @ 2.20GHz CPU and 377 GB RAM. The training and certification time for each method is reported in Table 12. We provide a detailed complexity analysis for each training method in Table 11.

Table 11: Detailed breakdown of training costs for each Certified Training method.

| Method | Training cost per batch | Details |
|---|---|---|
| Standard | T | Forward + Backward |
| PGD / EDAC | (M+1)T | M attack steps + Standard loss computation |
| IBP | 2T | Lower and Upper Bounds propagation |
| CROWN-IBP + LF | 4T | IBP pass + back-substitution of IBP bounds |
| SABR | (M+2)T | IBP + PGD |
| MTL-IBP | (M+2)T | IBP + PGD |
| TAPS | 2t + K*(M+1)*(T-t) | IBP for first network split and PGD for second split for each class |
| STAPS | 2t + K*(M+1)*(T-t) + (M+1)T | TAPS + PGD |

| | | Legend |
|---|---|---|
| | T | Time cost for Standard Training (Includes Forward + Backward Pass) |
| | M | Number of adversarial attack steps (including repeats) |
| | K | Number of classes |
| | t | Time cost for Standard Training in the first network split (TAPS) |

Table 12: Training and certification time for each method on different datasets and $\epsilon$.

| Dataset | $\epsilon$ | Method | Train Time (seconds) | Certification Time (seconds) |
|---------|-----------|--------|---------------------|------------------------------|
| MNIST | 0.1 | PGD | $1.5 \times 10^4$ | / |
| | | EDAC | $3.1 \times 10^4$ | / |
| | | IBP | $2.1 \times 10^3$ | $2.5 \times 10^3$ |
| | | CROWN-IBP | $5.6 \times 10^3$ | $1.8 \times 10^3$ |
| | | SABR | $1.8 \times 10^4$ | $6.0 \times 10^3$ |
| | | TAPS | $3.8 \times 10^4$ | $6.0 \times 10^3$ |
| | | STAPS | $2.5 \times 10^4$ | $6.9 \times 10^3$ |
| | | MTL-IBP | $6.8 \times 10^3$ | $6.8 \times 10^3$ |
| | 0.3 | PGD | $1.1 \times 10^4$ | / |
| | | EDAC | $2.2 \times 10^4$ | / |
| | | IBP | $2.6 \times 10^3$ | $3.2 \times 10^4$ |
| | | CROWN-IBP | $5.4 \times 10^3$ | $2.6 \times 10^4$ |
| | | SABR | $9.7 \times 10^3$ | $5.2 \times 10^4$ |
| | | TAPS | $7.1 \times 10^3$ | $4.7 \times 10^4$ |
| | | STAPS | $1.4 \times 10^4$ | $5.1 \times 10^4$ |
| | | MTL-IBP | $5.5 \times 10^3$ | $4.4 \times 10^4$ |
| CIFAR-10 | $\frac{2}{255}$ | PGD | $2.8 \times 10^4$ | / |
| | | EDAC | $1.3 \times 10^5$ | / |
| | | IBP | $1.2 \times 10^4$ | $1.3 \times 10^5$ |
| | | CROWN-IBP | $2.7 \times 10^4$ | $1.9 \times 10^5$ |
| | | SABR | $2.4 \times 10^4$ | $1.6 \times 10^5$ |
| | | TAPS | $1.1 \times 10^5$ | $1.1 \times 10^5$ |
| | | STAPS | $4.5 \times 10^4$ | $3.0 \times 10^5$ |
| | | MTL-IBP | $3.6 \times 10^4$ | $2.7 \times 10^5$ |
| | $\frac{8}{255}$ | PGD | $6.4 \times 10^4$ | / |
| | | EDAC | $1.3 \times 10^5$ | / |
| | | IBP | $1.1 \times 10^4$ | $1.9 \times 10^4$ |
| | | CROWN-IBP | $2.1 \times 10^4$ | $2.0 \times 10^4$ |
| | | SABR | $4.1 \times 10^4$ | $6.5 \times 10^4$ |
| | | TAPS | $3.3 \times 10^4$ | $4.0 \times 10^4$ |
| | | STAPS | $9.9 \times 10^4$ | $4.2 \times 10^4$ |
| | | MTL-IBP | $2.2 \times 10^4$ | $5.6 \times 10^4$ |
| TINYIMAGENET | $\frac{1}{255}$ | PGD | $1.0 \times 10^5$ | / |
| | | EDAC | $2.0 \times 10^5$ | / |
| | | IBP | $6.7 \times 10^4$ | $4.9 \times 10^3$ |
| | | CROWN-IBP | $2.0 \times 10^5$ | $1.3 \times 10^4$ |
| | | SABR | $1.1 \times 10^5$ | $1.8 \times 10^4$ |
| | | TAPS | $2.8 \times 10^5$ | $1.5 \times 10^4$ |
| | | STAPS | $3.3 \times 10^5$ | $2.6 \times 10^4$ |
| | | MTL-IBP | $1.5 \times 10^5$ | $5.1 \times 10^3$ |

# D. Additional Results

## D.1. Result Significance

In Tables 13–17, we present the natural and certified accuracy across certified training algorithms, datasets and perturbation levels. We report the average and standard deviation across 3 random seeds for each method. The results show that our improvements over previously reported values in the literature are significant, in most cases the statistical difference being larger than $3\sigma$.

Table 13: Comparison of Natural and Certified Accuracy between CTBENCH and previous literature results on MNIST $\epsilon = 0.1$. We report average and standard deviation across 3 random seeds for each method.

| Method | Source | Nat. [%] | Cert. [%] |
|---|---|---|---|
| IBP | Literature | 98.84 | 97.95 |
| | This work | $98.86 \pm 0.06$ | $98.25 \pm 0.03$ |
| CROWN-IBP | Literature | 98.83 | 97.76 |
| | This work | $98.93 \pm 0.01$ | $98.17 \pm 0.05$ |
| SABR | Literature | 99.23 | 98.22 |
| | This work | $99.15 \pm 0.08$ | $98.42 \pm 0.03$ |
| TAPS | Literature | 99.19 | 98.39 |
| | This work | $99.20 \pm 0.05$ | $98.5 \pm 0.04$ |
| STAPS | Literature | 99.15 | 98.37 |
| | This work | $99.15 \pm 0.04$ | $98.38 \pm 0.10$ |
| MTL-IBP | Literature | 99.25 | 98.38 |
| | This work | $99.16 \pm 0.03$ | $98.31 \pm 0.06$ |

Table 14: Comparison of Natural and Certified Accuracy between CTBENCH and previous literature results on MNIST $\epsilon = 0.3$. We report average and standard deviation across 3 random seeds for each method.

| Method | Source | Nat. [%] | Cert. [%] |
|---|---|---|---|
| IBP | Literature | 97.67 | 93.10 |
| | This work | $98.55 \pm 0.02$ | $93.82 \pm 0.10$ |
| CROWN-IBP | Literature | 98.18 | 92.98 |
| | This work | $98.46 \pm 0.03$ | $93.84 \pm 0.12$ |
| SABR | Literature | 98.75 | 93.40 |
| | This work | $98.69 \pm 0.03$ | $93.64 \pm 0.06$ |
| TAPS | Literature | 97.94 | 93.62 |
| | This work | $98.58 \pm 0.03$ | $93.90 \pm 0.11$ |
| STAPS | Literature | 98.53 | 93.51 |
| | This work | $98.69 \pm 0.06$ | $93.60 \pm 0.05$ |
| MTL-IBP | Literature | 98.80 | 93.62 |
| | This work | $98.75 \pm 0.02$ | $93.82 \pm 0.21$ |

Table 15: Comparison of Natural and Certified Accuracy between CTBENCH and previous literature results on CIFAR-10 $\epsilon = 2/255$. We report average and standard deviation across 3 random seeds for each method.

| Method | Source | Nat. [%] | Cert. [%] |
|---|---|---|---|
| IBP | Literature | 66.84 | 52.85 |
| | This work | $66.85 \pm 0.72$ | $55.32 \pm 0.68$ |
| CROWN-IBP | Literature | 71.52 | 53.97 |
| | This work | $67.56 \pm 0.04$ | $56.69 \pm 0.58$ |
| SABR | Literature | 79.24 | 62.84 |
| | This work | $77.82 \pm 0.28$ | $63.62 \pm 0.22$ |
| TAPS | Literature | 75.09 | 61.56 |
| | This work | $74.76 \pm 0.34$ | $61.37 \pm 0.09$ |
| STAPS | Literature | 79.76 | 62.98 |
| | This work | $76.88 \pm 0.15$ | $63.96 \pm 0.27$ |
| MTL-IBP | Literature | 80.11 | 63.24 |
| | This work | $78.91 \pm 0.16$ | $64.00 \pm 0.37$ |

Table 16: Comparison of Natural and Certified Accuracy between CTBENCH and previous literature results on CIFAR-10 $\epsilon = 8/255$. We report average and standard deviation across 3 random seeds for each method.

| Method | Source | Nat. [%] | Cert. [%] |
|---|---|---|---|
| IBP | Literature | 48.94 | 34.97 |
| | This work | $48.74 \pm 0.23$ | $34.99 \pm 0.28$ |
| CROWN-IBP | Literature | 46.29 | 33.38 |
| | This work | $48.24 \pm 0.09$ | $32.49 \pm 0.18$ |
| SABR | Literature | 52.38 | 35.13 |
| | This work | $52.51 \pm 0.38$ | $34.97 \pm 0.62$ |
| TAPS | Literature | 49.76 | 35.10 |
| | This work | $49.82 \pm 0.28$ | $34.89 \pm 0.40$ |
| STAPS | Literature | 52.82 | 34.65 |
| | This work | $51.46 \pm 0.25$ | $35.32 \pm 0.25$ |
| MTL-IBP | Literature | 53.35 | 35.44 |
| | This work | $53.72 \pm 0.49$ | $35.23 \pm 0.18$ |

Table 17: Comparison of Natural and Certified Accuracy between CTBENCH and previous literature results on TINYIMA-GENET $\epsilon = 1/255$. We report average and standard deviation across 3 random seeds for each method.

| Method | Source | Nat. [%] | Cert. [%] |
|---|---|---|---|
| IBP | Literature | 25.92 | 17.87 |
| | This work | $26.4 \pm 0.45$ | $19.87 \pm 0.19$ |
| CROWN-IBP | Literature | 25.62 | 17.93 |
| | This work | $28.16 \pm 0.27$ | $21.69 \pm 0.42$ |
| SABR | Literature | 28.85 | 20.46 |
| | This work | $30.96 \pm 0.41$ | $21.14 \pm 0.2$ |
| TAPS | Literature | 28.34 | 20.82 |
| | This work | $28.59 \pm 0.09$ | $21.54 \pm 0.22$ |
| STAPS | Literature | 28.98 | 22.16 |
| | This work | $30.25 \pm 0.33$ | $22.03 \pm 0.25$ |
| MTL-IBP | Literature | 37.56 | 26.09 |
| | This work | $35.97 \pm 0.17$ | $27.49 \pm 0.21$ |

## D.2. Architecture Generalization

In Table 18 we present the natural and certified accuracy on CNN5, which is a five-layer CNN smaller than CNN7. We observe that the improvements are consistent across different settings even with CNN5, showing that the improvements are not specific to a certain architecture.

Table 18: Comparison on CNN5 between CTBENCH and the implementation of De Palma et al. (2024).

| Method | Code and hyperparameters | Nat. [%] | Cert. [%] |
|---|---|---|---|
| IBP | CTBENCH | 98.19 | 92.88 |
| | (De Palma et al., 2024) | 93.16 | 81.81 |
| SABR | CTBENCH | 98.41 | 92.62 |
| | (De Palma et al., 2024) | 97.33 | 90.87 |
| MTL-IBP | CTBENCH | 98.41 | 92.49 |
| | (De Palma et al., 2024) | 98.39 | 91.45 |

## D.3. Ablation on Certification Algorithms & Additional Analysis on Shared Mistakes

In Table 19 we present the correlation between the certification capabilities of two SOTA verifiers (MN-BAB (Ferrari et al., 2022) and OVAL (De Palma et al., 2022)). We observe that there is a very high correlation between the two verifiers, which is expected since both are based on the same underlying principles. This shows that the certification algorithms have reached a certain level of maturity and are converging to similar results. Combining the verified sets of the two verifiers, we get a marginal improvement in certified accuracy at the cost of a much larger certification time. Therefore, we only use MN-BAB for certification in our main experiments.

Table 19: Observed count of common mistakes of certification algorithms (MN-BAB (Ferrari et al., 2022) and OVAL (De Palma et al., 2022)) on CIFAR-10 against their expected values assuming independence across certification mistakes.

|  |  | neither certify | one certifies | both certify |
|---|---|---|---|---|
| $\epsilon = 2/255$ | obs. | 3549 | 15 | 6436 |
|  | exp. | 1264 | 4585 | 4151 |
| $\epsilon = 8/255$ | obs. | 6454 | 9 | 3537 |
|  | exp. | 4171 | 4575 | 1254 |

In Table 20 we present the observed count of common mistakes that different certified training models make on CIFAR-10 against their expected values assuming independence across model mistakes. We observe that the observed count is significantly higher than the expected count, indicating that the models are highly correlated in their mistakes.

Table 20: Observed count of common mistakes on CIFAR-10 against their expected values assuming independence across model mistakes.

|  |  | # models succeeded | | | | | | |
|---|---|---|---|---|---|---|---|---|
|  |  | 0 | 1 | 2 | 3 | 4 | 5 | 6 |
| $\epsilon = \frac{2}{255}$ | obs. | 2350 | 653 | 520 | 564 | 708 | 894 | 4311 |
|  | exp. | 35 | 330 | 1296 | 2704 | 3163 | 1965 | 507 |
| $\epsilon = \frac{8}{255}$ | obs. | 5206 | 679 | 487 | 388 | 387 | 585 | 2268 |
|  | exp. | 766 | 2457 | 3283 | 2339 | 937 | 200 | 18 |

Furthermore, Table 21 examines shared mistakes between CNN5 and CNN7, revealing common patterns across architectures.

Table 21: Additional results on CNN5 and CNN7 shared mistakes for MNIST 0.3

| Models | Number of not certified samples | |
|---|---|---|
|  | Observed | Expected if independent |
| CNN5 IBP | 771 | / |
| CNN5 SABR | 793 | / |
| CNN5 MTL-IBP | 746 | / |
| CNN7 IBP | 620 | / |
| CNN7 SABR | 632 | / |
| CNN7 MTL-IBP | 610 | / |
| CNN5, CNN7 IBP | 526 | 48 |
| CNN5, CNN7 SABR | 541 | 50 |
| CNN5, CNN7 MTL-IBP | 516 | 46 |
| All 3 CNN5 networks | 593 | 5 |

## D.4. Deferred Results on CIFAR-10

In Figures 9 and 10, we present additional analyses on the neuron statistics for different models trained on CIFAR-10. We analyze the amount of unstable neurons and the model utilization for each model.

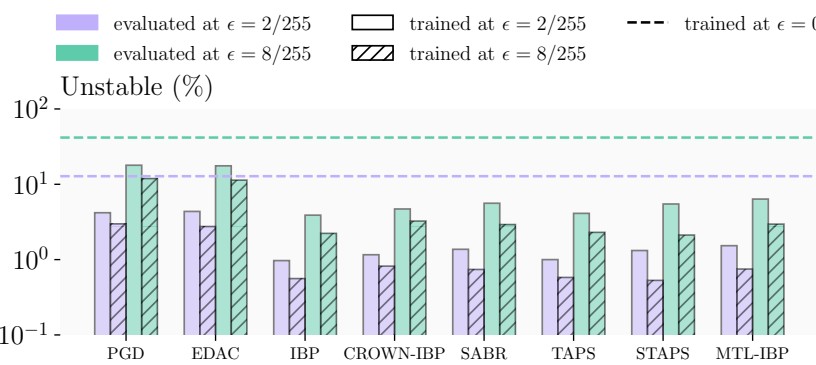

Figure 9: Ratio of unstable neurons for models trained on CIFAR-10 with different methods and $\epsilon$.

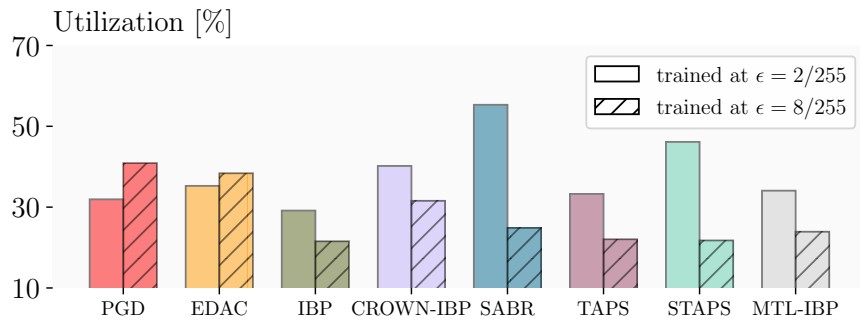

Figure 10: Model utilization for models trained on CIFAR-10 with different methods and $\epsilon$. We note that standard training has 35.79% utilization.

In Figure 11, we present the out-of-distribution generalization evaluated on CIFAR-10-C for models trained on CIFAR-10 at $\epsilon = 8/255$, $\epsilon = 2/255$ and standard training. We observe that the models trained with certified training methods have better out-of-distribution generalization compared to standard training.

### D.5. Results with Incomplete Certification Algorithms

In Table 22, we present the results of the certification with more efficient, but incomplete methods (IBP and CROWN-IBP). We observe that the incomplete methods have a significant impact on the certified accuracy, and the results are consistent with the previous findings in the literature.

Table 22: A comparison of incomplete certification (IBP and CROWN-IBP) against complete certification (MN-BAB) on CIFAR-10 $\epsilon = 2/255$.

| Train Method | Nat [%] | Cert [%] | | | Adv [%] |
| --- | --- | --- | --- | --- | --- |
| | | IBP | CROWN-IBP | MN-BAB | |
| IBP | 67.49 | **54.22** | 54.57 | 55.99 | 56.10 |
| CROWN-IBP | 67.60 | 49.92 | **54.86** | 57.11 | 57.28 |
| SABR | 77.86 | 12.12 | 44.79 | 63.61 | 65.56 |
| TAPS | 74.44 | 28.22 | 50.14 | 61.27 | 62.62 |
| STAPS | 77.05 | 0.72 | 30.92 | 64.21 | 66.09 |
| MTL-IBP | **78.82** | 0.62 | 23.31 | **64.41** | **67.69** |

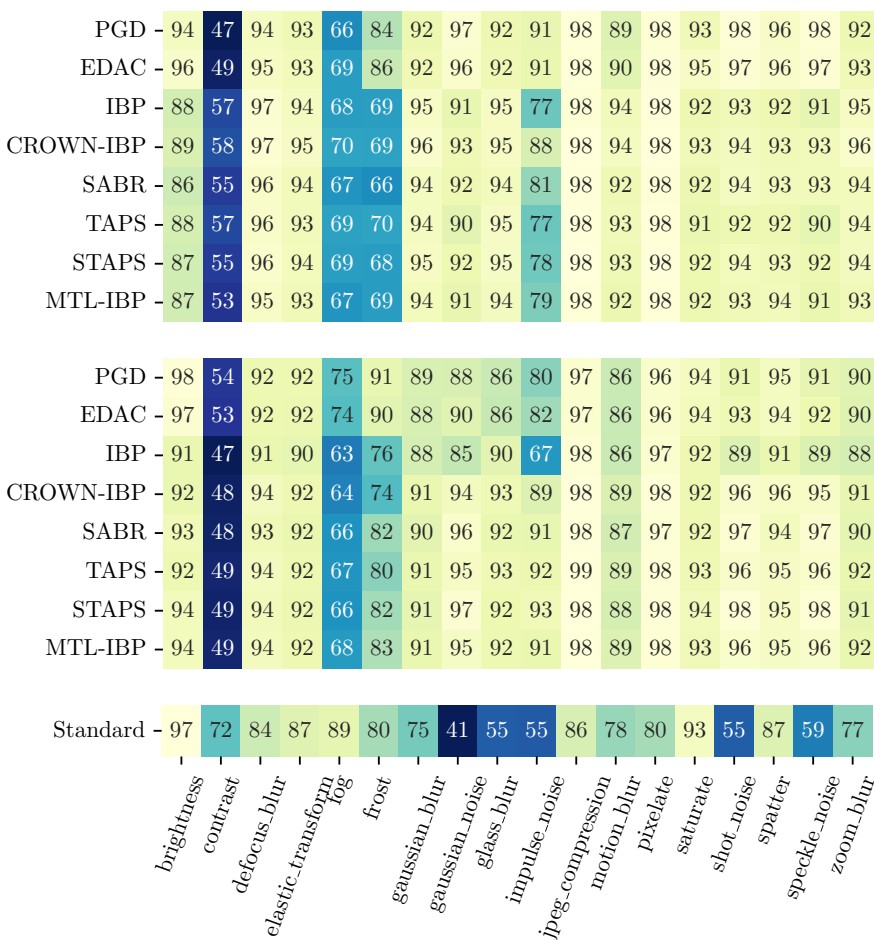

Figure 11: Out-of-distribution generalization evaluated on CIFAR-10-C for models trained on CIFAR-10 at $\epsilon = 8/255$ (top), $\epsilon = 2/255$ (middle) and standard training (bottom).

