# OpenReview forum: "CTBench: A Library and Benchmark for Certified Training"
_ICML.cc/2025/Conference — ICML 2025 poster_

### Official Review · Reviewer_VJvs · 2025-02-16

**Overall Recommendation:** 1

**Summary:**

While a number of algorithms for verifying the robustness of neural networks have been developed, it has also been shown that models trained using standard training approaches are often not robust and difficult to certify. Certified Training aims at developing methods which encourage verifiability and robustness during training while maintaining acceptable standard accuracy. This work introduces CTBench, a benchmark for comparing different certified training algorithms which were developed in recent years. The authors further include two adversarial training algorithms for completeness to enable a comparison of the different methods and their standard, empirical and certified accuracies. The authors further identify some issues in existing implementations of algorithm which they fix in their code base, and further unify the hyperparameter tuning methodology for different algorithms. The algorithms are evaluated on different standard datasets from the literature where the new implementation is found to lead to improved results in multiple cases. The analysis is complemented by a number of experiments on the effects that certified training has on the trained models.

**Claims And Evidence:**

- The experimental results only partially support the claims made by the authors. While the certified accuracy is improved by the authors' implementation in most cases, the improvements seem marginal in a number of cases.
The increase in robust accuracy is often accompanied by a decrease in standard accuracy which, in practical applications, would not be desirable (see e.g. the results for CIFAR10, $\epsilon=\frac{2}{255}$ for this). This issue is not sufficiently discussed by the authors and the discussion mostly focuses on the certified accuracies. Therefore, the experimental results don't seem to support statements such as "CTBench achieves consistent improvements in both certified and natural accuracies"
- The CTBench implementation of the algorithms shows improved performance in some cases. However, the authors change multiple aspects of the implementation (e.g. regarding BatchNorm statistics) and separately tune the methods' performance according to a specific scheme. There seem to be a number of moving parts in this work and it is not always clear to me which of the performance improvements are induced by which of the changes made by the authors. They discuss in the appendix that separating the changes is not always feasible, but in the current state of the paper this makes it hard to evaluate which changes make sense/have a positive effect. I assume that fixing mistakes in the implementation might also have effects on the way networks are evaluated at test time which makes it difficult to compare the results presented by the authors to previous results.

**Essential References Not Discussed:**

The authors state that "we only focus on $\ell_\infty$ robustness because there exists no deterministic certified training algorithm regarding other norms". Although the approach presented by [4] will struggle to scale to large network architectures and perturbations, it does explicitly address training for $\ell_2$ robustness, so this statement is incorrect. Since it should not be hard to extend bound propagation to other norms and since a number of works in verification do address robustness in these other norms, it would strengthen the paper if these were also considered.

[4] Soletskyi, R., & Dalrymple, D. (2024). Training Safe Neural Networks with Global SDP Bounds. arXiv preprint arXiv:2409.09687.

**Experimental Designs Or Analyses:**

The experimental setup and the hyperparameter tuning approach presented by the authors makes sense and I couldn't find any issues related to the experimental design.

**Methods And Evaluation Criteria:**

The authors include an evaluation of a number of state-of-the-art certified training algorithms in their evaluation. However, there are multiple aspects that are not considered by the authors, including e.g. probabilistic methods such as randomised smoothing or robustness to other norms such as $\ell_2$ or $\ell_1$. All of these are considered by the existing benchmark [1] but not discussed by the authors. Works such as [2] show that extending bound propagation to different norms is doable so the authors should extend their evaluation to also cover these other aspects when proposing a new benchmark.

[1] Li, L., Xie, T., & Li, B. (2023, May). Sok: Certified robustness for deep neural networks. In 2023 IEEE symposium on security and privacy (SP) (pp. 1289-1310). IEEE.

[2] Wang, Z., & Jha, S. (2023). Efficient symbolic reasoning for neural-network verification. arXiv preprint arXiv:2303.13588.

**Other Comments Or Suggestions:**

Small typo: Line 439 "training algorithms that explicitly has little loss fragmentation" --> training algorithms that explicitly **have** little loss fragmentation

**Other Strengths And Weaknesses:**

- The paper is quite polished and well-written, the explanations are clear
- Having a unified implementation of different algorithms will make it easier to benchmark new methods against existing ones

**Questions For Authors:**

- Do the authors have an intuition as to why certified training performs so poorly (and even worse than adversarial training, which is much less expensive) for some OOD corruptions?

## Post-rebuttal edit
Some points that should be addressed:
- The novelty of the work is severely limited, the paper provides an experimental evaluation which is similar to the evaluation section in any of the certified training papers that were published in recent years.
- Having a unified implementation seems useful, but the authors overstate the contributions that are made (e.g. in phrases such as "CTBench achieves consistent improvements in both certified and natural accuracies"), such statements should be toned down.
- Results on e.g. other networks or norms, or e.g. techniques such as randomised smoothing should be added to the paper to strengthen it

**Relation To Broader Scientific Literature:**

The paper basically implements a number of existing robust training algorithms in a common framework and fixes issues in the implementation of some of the algorithms. The novelty of the work is therefore quite limited. Section 5 analyses some other aspects of models trained with certified training algorithms, but some of the insights analysed are less surprising to readers familiar with the field. For example, the loss fragmentation discussed in section 5.1 seems closely related to the fact that certified training forces a number of neurons in the network to be either stably active or stably inactive as previously found by [3]. I also found the results on OOD generalisation to be somewhat difficult to parse since there seems to be no clear tendency as to when a specific certified training is actually helpful, which would make it quite difficult to decide which certified training algorithm to use in practice if improving OOD generalisation is an aim.

[3] Shi, Z., Wang, Y., Zhang, H., Yi, J., & Hsieh, C. J. (2021). Fast certified robust training with short warmup. Advances in Neural Information Processing Systems, 34, 18335-18349.

**Theoretical Claims:**

The paper has no theoretical claims which would require checking.

---

> ### Author Rebuttal · Authors · 2025-04-01
>
> We $\newcommand{\Rj}{\textcolor{purple}{VJvs}}$are happy to hear that Reviewer $\Rj$ finds our work interesting and well motivated, our library and benchmark useful and comprehensive, and our experimental results insightful. Due to the word limit, we address major questions raised by Reviewer $\Rj$ below and are happy to discuss more in the follow-up. We include new results as Table S1 etc., in the [anonymized link](https://mega.nz/file/eZp1CYBC#hwOYJzm4U47TDzuCmQFTJhCust2pFz-8Wzzy-CmXd6Q).
>
>
> **Q1: Could the authors comment on why the improvement of certified accuracy sometimes accompanies a decrease in natural accuracy?**
>
> The robustness-accuracy trade-off is well-known, where higher certified accuracy often comes at the cost of natural accuracy. Most methods, including SABR and MTL-IBP, have hyperparameters (e.g., $\lambda$, $\alpha$) that directly regulate this trade-off. Our goal, like in prior work, is to maximize certified accuracy, with natural accuracy improvements seen as a bonus. For completeness, we further provide robustness-accuracy curves, as shown in Figures S1–S3.
>
> **Q2: Could the authors provide an ablation study to evaluate the individual impact of CTBench strategies?**
>
> Please refer to our reply to Q4 of Reviewer $\textcolor{blue}{mj6P}$.
>
> **Q3: Can differences in implementation change the test-time certification?**
>
> No. Test-time certifications are conducted using third-party tools like MNBaB and OVAL. Our changes only affect the training process, while the final trained network can be exported and verified independently, ensuring direct comparability with the literature.
>
> **Q4: Could the authors discuss the connection between this work and probabilistic certification methods such as Randomised Smoothing?**
>
> Please refer to our reply to Q1 of Reviewer $\textcolor{green}{vyCo}$.
>
> **Q5: Could the authors discuss the possibility of extending this work to training and certification for other norms?**
>
> Our work focuses on deterministic certified training using bound propagation for the $L_\infty$ norm, as it remains the most reliable and widely adopted approach for robustness guarantees. While [1] explores various norms for certification, it also limits deterministic certified training to $L_\infty$, reflecting the current state of the field, with practical deterministic methods focused on $L_\infty$.
>
> Certification under other norms, such as $L_2$, faces scalability challenges. For example, [2] evaluates $L_2$ certification on small models with only 192 hidden nodes, while our CNN7 network has over 10M parameters, making their method impractical. Similarly, [4] uses expensive SDP methods, limiting their approach to synthetic toy datasets (Spheres) and does not naturally extend to $L_\infty$. Furthermore, while [4] addresses $L_2$-norm robustness, their methods do not naturally extend to $L_\infty$.
>
> We acknowledge that exploring deterministic certified training for other norms is a valuable future direction. However, due to scalability limitations and the lack of effective methods for other norms (even on MNIST), our focus remains on $L_\infty$. If Reviewer $\Rj$ knows of scalable methods for other norms, we would be happy to include them in our study. In addition, we will revise our statement to say that “we only focus on $L_\infty$ robustness because there exists no *scalable* deterministic certified training algorithm regarding other norms”.
>
> **Q6: Could the authors clarify the difference between the experiment with loss fragmentation in Section 5.1 and the findings of Shi et al. [3]?**
>
> We clarify that Shi et al. [3] analyze only IBP-based instability, which is an over-approximation of the real instability of neurons. In contrast, our analysis applies an estimate of the true number of unstable neurons. To illustrate this difference, we provide a comparison between the two variants in Table S5. We observe that the gap between our lower bound estimate and IBP is larger for SOTA methods, which also reflects in the certification gap between IBP and MN-BAB for these models (Table S4).
>
> **Q7: Do the authors have an intuition as to why certified training performs so poorly (and even worse than adversarial training, which is much less expensive) for some OOD corruptions?**
>
> On corrupted datasets (MNIST-C, CIFAR-10-C), adversarial and certified training improve robustness against localized perturbations like blur, noise, and pixelation but struggle with global shifts like brightness and contrast changes. This aligns with the intuition that these methods enhance robustness mainly in the immediate neighborhood of the original inputs, whereas global changes fall outside this region. Moreover, the stronger regularization induced by certified training when compared to adversarial training often exceeds what's needed for untargeted corruptions. Addressing this may require diverse augmentations, though it could reduce certified adversarial accuracy.

---

### Official Review · Reviewer_mj6P · 2025-03-07

**Overall Recommendation:** 3

**Summary:**

This paper introduces a novel benchmark for certified training, addressing the inconsistencies in evaluating certifiably robust neural networks. Existing methods suffer from unfair comparisons due to varying training schedules, certification techniques, and under-tuned hyperparameters, leading to misleading claims of improvement. CTBench standardizes evaluations by integrating state-of-the-art certified training algorithms into a single codebase, systematically tuning hyperparameters, and correcting implementation issues, thus reestablishing a stronger state-of-the-art. The study reveals several key insights: (1) certified models have less fragmented loss surface, (2) certified models share many mistakes, (3) certified models have more sparse activations, (4) reducing regularization cleverly is crucial for
certified training especially for large radii, and (5) certified training has the potential to improve out-of-distribution generalization.

## update after rebuttal
The rebuttal helped improve the soundness of empirical results. Hence, I preserve my positive recommendation.

**Claims And Evidence:**

This paper proposes a novel benchmark for certified training, and its claims and insights are primarily supported by experimental results conducted within this benchmark. While the findings are well-documented, one notable concern is that all results are derived from a single CNN7 architecture, limiting the generalizability of the conclusions. Including more diverse network architectures would strengthen the validity of the results.

**Essential References Not Discussed:**

I don't see any missing references.

**Experimental Designs Or Analyses:**

Most of the experimental designs make sense to me. Here are some of my concerns:

1. CTBench uses the number of unstable neurons to represent the smoothness of the loss surface. This can pose difficulty in generalizing to non-ReLU neural networks such as Swish [1] and GELU [2]. A more explicit way is to focus on the change of the loss value within a neighborhood of an input sample.

2. CTBench achieves a state-of-the-art certified training performance. To achieve this desirable performance, several strategies are taken, such as batch norm, hyperparameter tuning, and L1 regularization. It would be helpful to add an ablation study on these strategies.

3. Complexity is an important metric for certified training. However, the running time is only provided in the appendix (Table 8). It would be helpful to add more analyses and discussions on the complexity of different certifying methods.

4. The experiment of shared mistakes does not provide a general insight. The result is specific to the data. It is unclear to me why curriculum learning can improve certified training based on the results.

[1] Ramachandran, Prajit, Barret Zoph, and Quoc V. Le. "Searching for activation functions." arXiv preprint arXiv:1710.05941 (2017).

[2] Hendrycks, Dan, and Kevin Gimpel. "Gaussian error linear units (gelus)." arXiv preprint arXiv:1606.08415 (2016).

**Methods And Evaluation Criteria:**

Yes. The evaluation metrics follow the previous studies on certified training, including certified accuracy, natural accuracy, and adversarial accuracy.

**Other Comments Or Suggestions:**

Please see Strengths And Weaknesses

**Other Strengths And Weaknesses:**

Strengths:
- This paper proposes a comprehensive benchmark for certified training. This benchmark standardizes the evaluation of deterministic certified training methods, enabling fair comparisons.
- Based on the proposed CTBench, this paper achieves state-of-the-art certified training performance by carefully adapting previous methods.
- This paper also reveals new insights from the evaluation results of existing methods.
- This paper is overall well organized and easy to follow.

Weaknesses: please see my concerns in the Claims And Evidence and Experimental Designs Or Analyses sections. I list them below:
- While the findings are well-documented, one notable concern is that all results are derived from a single CNN7 architecture, limiting the generalizability of the conclusions. Including more diverse network architectures would strengthen the validity of the results.
- CTBench uses the number of unstable neurons to represent the smoothness of the loss surface. This can pose difficulty in generalizing to non-ReLU neural networks such as Swish [1] and GELU [2]. A more explicit way is to focus on the change of the loss value within a neighborhood of an input sample.
- CTBench achieves a state-of-the-art certified training performance. To achieve this desirable performance, several strategies are taken, such as batch norm, hyperparameter tuning, and L1 regularization. It would be helpful to add an ablation study on these strategies.
- Complexity is an important metric for certified training. However, the running time is only provided in the appendix (Table 8). It would be helpful to add more analyses and discussions on the complexity of different certifying methods.
- The experiment of shared mistakes does not provide a general insight. The result is specific to the data. It is unclear to me why curriculum learning can improve certified training based on the results.

[1] Ramachandran, Prajit, Barret Zoph, and Quoc V. Le. "Searching for activation functions." arXiv preprint arXiv:1710.05941 (2017).

[2] Hendrycks, Dan, and Kevin Gimpel. "Gaussian error linear units (gelus)." arXiv preprint arXiv:1606.08415 (2016).

**Questions For Authors:**

Please see Strengths And Weaknesses

**Relation To Broader Scientific Literature:**

This paper contributes to the broader literature by standardizing the evaluation of certified training methods. While prior works have introduced individual certified training algorithms, there has been no unified benchmark to fairly compare these methods under standardized conditions. By providing a standardized benchmark and implementation framework, this paper ensures that future studies can fairly compare new certified training methods against a well-tuned set of baselines, preventing misleading claims due to unfair comparisons. This aligns with broader machine learning efforts in reproducibility, benchmarking, and robustness evaluation.

**Theoretical Claims:**

No theoretical claims are made in the paper.

---

> ### Author Rebuttal · Authors · 2025-04-01
>
> We $\newcommand{\Rm}{\textcolor{blue}{mj6P}}$thank Reviewer $\Rm$ for their insightful review. We are happy to hear that Reviewer $\Rm$ finds our work interesting and well motivated, our library and benchmark useful and comprehensive, and our experimental results insightful. In the following, we address all concrete questions raised by Reviewer $\Rm$. We include new results as Table S1 etc., in the [anonymized link](https://mega.nz/file/eZp1CYBC#hwOYJzm4U47TDzuCmQFTJhCust2pFz-8Wzzy-CmXd6Q).
>
> **Q1: Can the findings of this study be generalised to other architectures than CNN7?**
>
> Our study focuses on the CNN7 architecture, which is consistently adopted by SOTA works to enable direct comparison. Table 14 in Appendix C2 compares CTBench training with the baseline training code from [1] using a smaller CNN5 on MNIST with $\epsilon = 0.3$. Results show that CTBench improves IBP, SABR, and MTL-IBP performance on CNN5, proving that the benefits extend beyond CNN7. Moreover, it is confirmed on CNN5 that IBP can match SOTA methods under large perturbations with fair comparisons and well-chosen hyperparameters.
>
> Furthermore, Table S2 examines shared mistakes between CNN5 and CNN7, revealing common patterns across architectures. We will include additional results on generalizability in the revised manuscript.
>
> **Q2: The number of unstable neurons as the metric of smoothness may be challenging to generalize to non-ReLU networks. Could the authors comment on this and consider alternatives, such as measuring the change in loss value within a neighborhood of an input sample?**
>
> We agree that the number of unstable neurons is hard to generalize to non-ReLU networks, as a metric of smoothness. However, it is of critical importance to the certification of ReLU networks, beyond measuring the difficulty of adversarial attacks which may also be indicated by other smoothness metrics. Concretely, branch-and-bound (BaB), the dominating strategy for complete certification of ReLU networks, directly branches the unstable neurons; thus, the number of unstable neurons provides a direct metric for the difficulty of certification. Since ReLU networks dominate certified training, we adopt the number of unstable neurons as the main metric. We will clarify this and comment on generalization to non-ReLU networks in the revised manuscript.
>
> **Q3: CTBench achieves SOTA performance by incorporating different strategies. Could the authors discuss the individual impact of these strategies?**
>
> We acknowledge the importance of ablation studies and discuss in Appendix A the challenge of fully disentangling these effects due to their interconnected nature.
>
> While full disentanglement is infeasible, we conduct a preliminary study to separate implementation advantages from hyperparameter tuning. Table 14 compares CNN5 performance using CTBench and the SOTA codebase, applying CNN7-tuned hyperparameters to both to reduce tuning bias, showing CTBench's universal implementation benefits. Additionally, Table S3 (L1 regularization on IBP-trained networks) and Figure S3 (effects of varying $\lambda$ for SABR and STAPS and $\alpha$ for MTL-IBP on the robustness-accuracy trade-off) illustrate hyperparameter impact.
>
> **Q4: Complexity is a crucial metric, but the running time is only reported in the appendix. Could the authors provide more analysis and discussion on the complexity of different training and certifying methods?**
>
> We agree that complexity is crucial. Therefore, in addition to the running times reported in Appendix (Table 8), we provide a more detailed complexity analysis for each training method in Table S6.
>
> For certification, we use complete certification algorithms based on branch-and-bound techniques that have exponential complexity, depending on the number of unstable neurons that need to be analyzed. Due to this exponential growth, we use a timeout of 1000 s per sample. Also, we report a new ablation study on certification algorithms in Table S4, including IBP and CROWN-IBP (efficient but incomplete certification algorithms) certified accuracies.
>
> **Q5: The experiment on shared mistakes appears to provide dataset-specific results rather than general insights. Could the authors clarify why curriculum learning improves certified training?**
>
> We would like to clarify that we do not claim curriculum learning directly improves certified training. Our claim is that data points exhibit varying levels of difficulty for certified training, as evidenced by the systematic deviation from independent errors. The connection between data point difficulty and curriculum learning is discussed in [2], which is why we suggest its potential benefit. Additionally, Appendix C.3 provides a more extensive evaluation across different datasets and certification algorithms, demonstrating that the results are not specific to a single dataset.
>
> **References**
>
> [1] arxiv.org/abs/2305.13991
>
> [2] arxiv.org/abs/1705.08280

---

### Official Review · Reviewer_vyCo · 2025-03-11

**Overall Recommendation:** 4

**Summary:**

The authors proposed to do a new round of meta-research on the topic of certified training (because the previous one [1] became outdated), compared the top algorithms and baselines with a fair training pipeline, and are going to share a library and related benchmark for further usage.

[1] Linyi Li, Tao Xie, and Bo Li. Sok: Certified robustness for deep neural networks. In SP, pp. 1289–1310. IEEE, 2023.

## update after rebuttal
Authors provided more analysis on the diverse datasets so I will keep my original score.

**Claims And Evidence:**

Main (meta) claims:
1. almost all algorithms in CTBENCH surpass the corresponding reported performance in literature in the magnitude of algorithmic improvements, thus establishing new state-of-the-art, and
2. the claimed advantage of recent algorithms drops significantly when we enhance the outdated baselines with a fair training schedule, a fair certification method and well-tuned hyperparameters.

Additionally, the following insights are provided:
1. certified models have less fragmented loss surface,
2. certified models share many mistakes,
3. certified models have more sparse activations,
4. reducing regularization cleverly is crucial for certified training especially for large radii and
5. certified training has the potential to improve out-of-distribution generalization.

Actually, all of the claims are supported by careful experimentation pipeline and analysis.

**Essential References Not Discussed:**

No

**Experimental Designs Or Analyses:**

A very good experimentation part - actually, the whole paper is one big careful experimentation pipeline.

**Methods And Evaluation Criteria:**

Although the datasets have the low resolution (like 32x32), taking into account the difficulty of formal verification for deep neural nets, it's the trade-off between realistic data/NN and meaningful performance.

**Other Comments Or Suggestions:**

No

**Other Strengths And Weaknesses:**

In addition to meta-research on randomized certified robustness, I think the following this is missing in the paper: there are a lot of comparisons of methods and other finer grained details *inside* one dataset, but almost zero comparisons of the differences of performance and trends on different datasets - like OOD performance difference on MNIST-C vs CIFAR-10-C vs TinyImageNet, etc. Such sort of insights would be very important to have a meta-view on the datasets as well, not only algorithms.

**Questions For Authors:**

No

**Relation To Broader Scientific Literature:**

The work is related to a more broad certified area of research that comprises both deterministic certified training (the scope of the reviewed paper) and randomized certified robustness [1]. It would be really great to provide a meta-research in this area as well, because highly likely the problems with unfair comparison do exist there as well.

[1] Jeremy M. Cohen, Elan Rosenfeld, and J. Zico Kolter. Certified adversarial robustness via randomized smoothing. In Proc. of ICML, 2019.

**Theoretical Claims:**

No really theoretic concepts were covered in the paper. There were some hypotheses that were to some extent proved by experiments (like fragmented loss surface and others) but nothing more.

---

> ### Author Rebuttal · Authors · 2025-04-01
>
> We $\newcommand{\Rv}{\textcolor{green}{vyCo}}$thank Reviewer $\Rv$ for their insightful and careful review. We are happy to hear that Reviewer $\Rv$ finds our work interesting and well motivated, our experimental results convincing, and our ablation studies insightful. In the following, we address all concrete questions raised by Reviewer $\Rv$. We include new results, named with Table S1 etc., in the [anonymized link](https://mega.nz/file/eZp1CYBC#hwOYJzm4U47TDzuCmQFTJhCust2pFz-8Wzzy-CmXd6Q).
>
> **Q1: How are the issues of unfair comparison highlighted in this work relate to the broader area of certified robustness, namely non-deterministic methods such as randomized smoothing (RS)?**
>
> We agree that unfair comparison may also be present in the RS literature. After a brief survey, we find that recently published randomized smoothing techniques vary significantly in network architecture, training schedules, hyperparameter choices, and the noise distributions used for certification, all of which may contribute to unfair comparisons. Therefore, introducing a standardized fair benchmark as well as unified implementation for RS is also important. However, while a meta-survey on the issue may be easily performed, establishing a benchmark at the same level of CTBench requires a large amount of effort, and is way beyond the scope of this work. Concretely, one needs to unify all algorithms in a single library (which has never been developed, to the best of our knowledge), validate that all implementations match (or exceed, as in CTBench) the original reports, and then may start to evaluate them in fair settings. We remark that even the last step, which requires the least human effort, takes months’ compute on four GPUs in CTBench. Therefore, it is impractical for us to conduct the same study for RS in this work. However, we strongly believe that this is a good future work, and we will discuss this direction and the meta-survey in the revised manuscript.
>
> Meanwhile, we conduct a preliminary study on comparing $L_\infty$-norm robustness certified by RS to our results based on deterministic algorithms. Specifically, we compare the numbers by the state-of-the-art $L_\infty$-norm RS algorithm [1] on CIFAR-10 $\epsilon=2/255$ and $\epsilon=8/255$ with CTBench results in Table S1. We find that the current RS approaches yield lower certified accuracy compared to CTBench, in agreement with the literature that deterministic methods dominate the $L_\infty$ robustness.
>
> **Q2: The paper includes results for multiple datasets, but each of them is analysed separately. Could the authors provide a further analysis and comparison of performance trends across the datasets?**
>
> Sure, we provide a preliminary analysis below, and will include more in the revised manuscript.
>
> Across the datasets considered in this work, several performance trends emerge, offering insights into how different certification and training methods generalize. For both MNIST and CIFAR-10, we observe that Interval Bound Propagation (IBP) demonstrates great performance at larger perturbation sizes, while other methods show limited improvement over IBP. This suggests that as perturbations increase in magnitude, stronger regularization is crucial for maintaining certifiability. In the context of corrupted datasets (MNIST-C and CIFAR-10-C), adversarial and certified training methods effectively enhance robustness against localized perturbations such as blur, noise, and pixelation. However, these methods remain less resilient to global transformations like brightness and contrast changes compared to standard training. This observation aligns with the intuition that adversarial and certified training primarily improve robustness in the immediate neighborhood of the original inputs, whereas global changes fall outside this region. Addressing this limitation could involve more diverse data augmentation strategies, though this may come at the cost of reduced certified adversarial accuracy.
>
> When examining network-level properties such as neuron instability and network utilization, trends across datasets are less straightforward. In all cases, standard training results in the highest neuron instability, as expected due to the absence of regularization aimed at minimizing this effect. However, network utilization does not follow a consistent pattern. In some scenarios, certified training increases network utilization compared to adversarial training, indicating the learning of more complex patterns and relationships. Yet this trend is not universally observed, suggesting that the underlying dynamics of network utilization are context-specific and not easily generalizable.
>
> Overall, these findings highlight that while some performance trends persist across datasets, others are context-dependent, underscoring the need for context-specific analysis when evaluating the current stage of certified robustness methods.
>
> **References**
>
> [1] arxiv.org/abs/2406.10427

---

> > ### Comment · Reviewer_vyCo · 2025-04-02
> >
> > Thanks authors for answering my remarks! I'll appreciate more diverse analysis on different datasets used to be included into the final text.
> >
> > And I do believe / hope on the final meta-analysis for RS. Now it is a zoo.

---

### Decision · Program_Chairs · 2025-05-01

**Decision:**

Accept (poster)

**Comment:**

This paper introduces CTBench, a unified library and benchmark for deterministic certified training of neural networks. CTBench re-implements existing certified training algorithms under a shared codebase with systematically tuned hyperparameters, enabling fair and consistent evaluation. The authors observe that, under this standardized setting, existing algorithms perform better than originally reported. Moreover, the improvements claimed by recent methods diminish when compared against baselines with carefully tuned training schedules. The paper also offers insights into certified training based on these findings.

All three reviewers appreciate the well-designed experiment pipeline, the contribution of having a unified implementation to compare different algorithms, and the well-organized experiment observations and the well-written paper. However, reviewers also raised concerns about the missing evaluations of randomized smoothing methods (reviewers vyCo, VJvs), limited generalizability due to the use of a single architecture (reviewer mj6P), missing general insights across different datasets (reviewers vyCo, mj6P), limited novelty beyond evaluation of existing algorithms (reviewer VJvs), and missing ablation study and discussion about training complexity (reviewer mj6P).

During the rebuttal phase, the authors provided additional experiments and analyses to address some of the reviewers’ concerns, including the preliminary study on randomized smoothing, the results of another architecture, the impacts of hyperparameter tuning, the training complexity, and the analyses across datasets. The reviewers acknowledged that these results addressed their original concerns. However, some concerns still remain, such as the limited novelty and overstated contributions. Considering all reviews and given the contribution of a unified evaluation pipeline, the AC recommends weak acceptance for the paper.